# Molecular transitions in early progenitors during human cord blood hematopoiesis

Shiwei Zheng[1,2], Efthymia Papalexi[1,2], Andrew Butler[1,2], William Stephenson[1] & Rahul Satija[1,2,*] 

## Abstract

Hematopoietic stem cells (HSCs) give rise to diverse cell types in the blood system, yet our molecular understanding of the early trajectories that generate this enormous diversity in humans remains incomplete. Here, we leverage Drop-seq, a massively parallel single-cell RNA sequencing (scRNA-seq) approach, to individually profile 20,000 progenitor cells from human cord blood, without prior enrichment or depletion for individual lineages based on surface markers. Our data reveal a transcriptional compendium of progenitor states in human cord blood, representing four committed lineages downstream from HSC, alongside the transcriptional dynamics underlying fate commitment. We identify intermediate stages that simultaneously co-express "primed" programs for multiple downstream lineages, and also observe striking heterogeneity in the early molecular transitions between myeloid subsets. Integrating our data with a recently published scRNA-seq dataset from human bone marrow, we illustrate the molecular similarity between these two commonly used systems and further explore the chromatin dynamics of "primed" transcriptional programs based on ATAC-seq. Finally, we demonstrate that Drop-seq data can be utilized to identify new heterogeneous surface markers of cell state that correlate with functional output.

**Keywords** hematopoiesis; single cells; single-cell RNA-seq; transcriptional dynamics
**Subject Categories** Development & Differentiation; Genome-Scale & Integrative Biology; Transcription
**Mol Syst Biol. (2018) 14: e8041**

## Introduction

Hematopoiesis is the dynamic process by which a single hematopoietic stem cell (HSC) can give rise to the breathtaking cellular diversity present in blood, potentially representing tens to hundreds of distinct cell types which can be loosely grouped into erythroid, myeloid, and lymphoid lineages (Becker *et al*, 1963; Orkin, 2000; Orkin & Zon, 2008; Seita & Weissman, 2010). However, despite enormous biological and clinical relevance, the molecular trajectories that cells traverse during lineage commitment remain poorly understood. Seminal experimental work in the mouse has suggested a model where individual HSCs undergo a sequential loss of pluripotency and pass through distinct intermediate progenitors represented by a series of binary branchings, with the first lineage decision representing either myelo-erythroid or lymphoid specification (Kondo *et al*, 1997; Akashi *et al*, 2000).

Recent studies, however, have proposed both minor and major alterations to the structure of the traditional model, for example positing a direct path from HSC to erythroid and megakaryocytic lineages (Adolfsson *et al*, 2005) or demonstrating diverse lineage origins for myeloid cells (Franco *et al*, 2010; Drissen *et al*, 2016), all highlighting a lack of consensus regarding the molecular nature of early fate transitions in hematopoiesis. The evidence for each of these models is based primarily on the enrichment of putative progenitor cell populations from fluorescence-activated cell sorting (FACS). Even slight differences in the surface markers utilized, gating strategy for enrichment, or downstream assay conditions can skew the output and interpretation of these experiments (Etzrodt *et al*, 2014; Paul *et al*, 2015). Moreover, the protocols used to identify intermediate progenitor types can vary widely between different laboratories, necessitating unsupervised approaches to define transition states at the single cell level (Levine *et al*, 2015; Nestorowa *et al*, 2016). This is particularly true in human hematopoiesis, as well-characterized markers in mouse (i.e., *Sca-1*) do not directly translate to human systems (Doulatov *et al*, 2012).

By contrast, single-cell RNA-seq (scRNA-seq) can provide a detailed molecular characterization of single cells that is highly complementary to traditional differentiation or FACS-based phenotyping approaches (Chattopadhyay *et al*, 2014). Massively parallel approaches that barcode cells in early stages of library preparation have enabled the routine profiling of thousands of single cells (Jaitin *et al*, 2014; Klein *et al*, 2015; Macosko *et al*, 2015), and the computational reconstruction of complex developmental processes (Haghverdi *et al*, 2016; Setty *et al*, 2016; Qiu *et al*, 2017). For example, a massively parallel scRNA-seq study of thousands of myeloid-restricted cells from the mouse bone marrow poignantly demonstrated that individual cells in this pool (which was depleted for the early progenitors expressing stem cell marker *Sca-1*) had largely committed to individual lineages (Paul *et al*, 2015). Additionally, a recent pioneering study of human bone marrow CD34[+] cells combining single-cell transcriptional and functional analysis (Velten *et al*, 2017) highlighted the continuous nature of early hematopoietic

1 New York Genome Center, New York, NY, USA
2 Center for Genomics and Systems Biology, New York University, New York, NY, USA
*Corresponding author. Tel: +1 646 977 7000; E-mail: rsatija@nygenome.org

differentiation and concluded that lineage commitment was not characterized by distinct branching during early transitions. In particular, the study focused on the presence of a "cloud-like" population of early multipotent progenitors ("CLOUD") forming an interconnected hierarchy, which then gave rise to distinct lineage-committed populations.

These complementary datasets and findings suggest that oligopotent progenitors may play a reduced role compared to initially proposed hierarchical models, but raise a pressing question for human hematopoiesis: Can all progenitors be stratified into either fully uncommitted or unipotent populations? Additionally, is there molecular evidence for multilineage priming in early progenitors, perhaps evidenced by co-expression of multiple genes in early cells that later become restricted to a single lineage? A deeper exploration of these questions could help bridge the insights derived from scRNA-seq and complementary techniques, including *in vivo* barcoding assays and both *in vivo* and *in vitro* differentiation experiments, all of which reveal evidence for oligopotent states, albeit with non-uniform lineage outputs (Doulatov *et al*, 2010; Kohn *et al*, 2012; Naik *et al*, 2013; Lee *et al*, 2017; Pei *et al*, 2017). In particular, recent work suggested that human umbilical cord blood could be a particularly attractive model to study, as this system contains a greater percentage of cells in oligopotent intermediate states compared to adult bone marrow (Notta *et al*, 2016).

In this study, we applied Drop-seq to individually sequence 21,306 stem and progenitor cells from umbilical cord blood, representing five healthy donors. Our data revealed the presence of single-cell clusters whose expression profiles were in tight agreement with previously defined progenitor populations, in addition to the presence of distinct myeloid progenitor subtypes. Importantly, we also observed a continuum of transitioning states that link these progenitor groups, enabling us to computationally reconstruct molecular trajectories connecting HSCs to four hematopoietic lineages. Our scRNA-seq data revealed strong evidence for recently proposed models that human myeloid cells can arise from distinct trajectories, with a subset of granulocytes sharing early molecular transitions with erythroid progenitors. We also observed that early progenitors co-expressed "primed" expression programs, associated with the commitment to multiple downstream lineages, indicating multilineage priming within intermediate stages. By integrating scRNA-seq datasets from cord blood and bone marrow, we further demonstrated strong molecular conservation in both systems and identify epigenetic trends that correlate with transcriptomic dynamics using a bone marrow bulk ATAC-seq dataset. Finally, we show that by coupling scRNA-seq with immunophenotyping measurements, Drop-seq data can be used to suggest new heterogeneous markers of cell state and potential, which we validate through *in vitro* differentiation assays. Our results shed new light on the molecular nature of early fate transitions in human hematopoiesis and highlight the exciting potential for high-throughput single-cell analysis to deconvolve complex developmental systems.

# Results

### Unsupervised identification of cellular diversity in human CD34⁺ cord blood cells

In order to characterize cellular heterogeneity at early stages of human hematopoiesis, we applied a recently developed massively

parallel single-cell library preparation technique, Drop-seq (Macosko *et al*, 2015), to sequence progenitor cells from human cord blood samples. The cord blood CD34$^+$ pool has been widely accepted as a rich source of hematopoietic stem and progenitor cells (Broxmeyer *et al*, 1989; Gluckman *et al*, 1989; Nimgaonkar *et al*, 1995), and we therefore sought to minimize our sample preparation steps prior to single-cell profiling, using enriched CD34$^+$ cells from five umbilical cord blood units with density centrifugation and magnetic separation (Materials and Methods; Fig 1A). In Drop-seq, single cells were co-encapsulated with barcoded beads in oil-based droplets, followed by pooled library preparation, sequencing, read alignment, and gene quantification based on unique molecular identifiers (UMIs), as previously described (Macosko *et al*, 2015). We selected input concentration for cells and micro-particles, as well as their corresponding flow rates during the microfluidic runs (Materials and Methods), to aim for a doublet rate of 1–2% according to human–mouse species-mixing experiments. Overall, our dataset contains 21,306 single cells across five biological replicates after initial filtering based on technical metrics (Materials and Methods), with an average of 27,513 mapped reads/cell. In total, 30,730 genes were detected across all cells, and 1,046 genes and 2,154 UMIs were assigned to each cell on average. We applied a latent variable model to control for cell cycle effects and technical covariates in our data (Buettner *et al*, 2015).

We next sought to identify the transcriptional subtypes and states comprising the CD34$^+$ progenitor pool. We extended our previously developed clustering strategy from Drop-seq data (Macosko *et al*, 2015; Satija *et al*, 2015) to reveal the presence of transcriptionally defined cellular subpopulations, across a wide range of abundances (15–0.4%). Briefly, we determined clusters by first reducing the data to 24 independent components, and then identified distinct clusters using a community detection strategy that has recently been applied to CyTOF data (Levine *et al*, 2015; Materials and Methods). We identified a few rare clusters that likely represented CD34$^{low/-}$ cells that passed through our column, including $CD3^+$ T, $KLRB1^+$ NK cells, $MS4A1^+$ B cells, and $C5AR1^+$ myeloid cells, and excluded these cells from additional analyses (Fig EV1A and B). Figure 1B shows a heat map of the remaining 19,394 cells, representing ten clusters, alongside the strongest transcriptional markers for each subpopulation (Materials and Methods; Table EV1). We further verified that our clustering results are consistent across a range of parameter settings, by repeating single-cell clustering over paired combinations of five resolution parameters and five nearest-neighbor numbers, creating 25 clusterings in total (Materials and Methods). As shown in Fig EV1C, pairs of cells that clustered together in the original analyses consistently clustered together across parameter settings. We therefore conclude that our original clustering faithfully represents our single-cell data and is not tuned to particular parameter values. The percentage of cells assigned to each cluster was highly consistent across all five biological replicates (Fig 1D and Table EV2; average $R = 0.94$), demonstrating that our five independent cord blood units spanned the same range of gene expression states, and at similar densities.

We next sought to understand if the clusters we identified from the total CD34$^+$ population contained subpopulations that were consistent with well-described progenitor populations. We compared our data to a recently published microarray reference dataset, containing bulk expression profiles for sorted populations

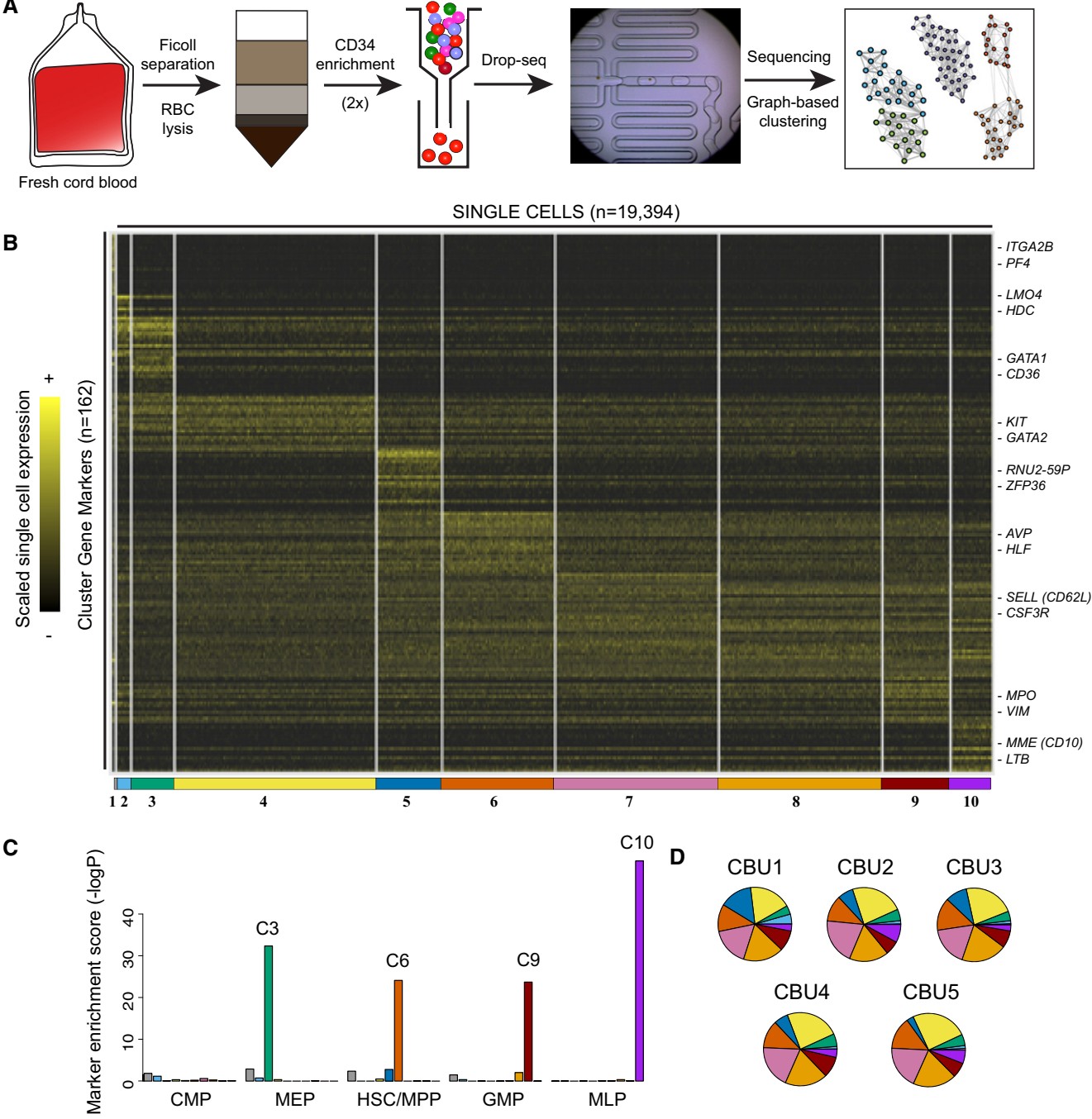

**Figure 1.  Identification of progenitor states in human cord blood hematopoiesis.**

A  Schematic of experimental workflow, consisting of column-based CD34 enrichment from human cord blood mononuclear cells, followed by Drop-seq.

B  Single-cell heatmap of 19,394 cord blood progenitors. Shown are ten progenitor states identified from graph-based clustering, and their strongest transcriptional markers. For visualization, expression for each gene is scaled (z-scored) across single cells.

C  Marker enrichment tests (Materials and Methods) comparing Drop-seq clusters and reference cell types from Laurenti *et al* (2013). Our unbiased clustering recovered well-characterized progenitor states, but we did not observe a cluster consistent with a traditional common myeloid progenitor (CMP).

D  Compositional makeup of five independent cord blood units (CBUs). The width and color of each slice correspond to the percentage of cells in each CBU represented in each cluster.

(Laurenti *et al*, 2013; Materials and Methods). Strikingly, we observed near-perfect overlap for many of our clusters with these profiles, enabling us to assign putative biological identities (Fig 1C).

For example, cluster 3 (C3) was characterized by erythroid fate regulators and markers *GATA1*, *CD36*, and *KLF1* (Pevny *et al*, 1991; van Schravendijk *et al*, 1992) and the entire set of markers we

identified in C3 were highly enriched for expression in the MEP reference microarray profiles (Fig 1C). Similarly, we assigned C6 to an HSC/multipotent progenitor (MPP) identity, C9 to the granulo-cyte/macrophage progenitor (GMP), and C10 to the multilymphoid progenitor (MLP). We also identified a second cluster similar to HSC (C5), expressing similar markers as well as *ZFP36* and a set of small RNAs, potentially representing HSC in a different metabolic state (Cheung & Rando, 2013). We did not, however, discover a cluster whose gene expression patterns were consistent with a common myeloid progenitor (CMP) state. This observation is consistent with the possibility that the currently defined human CMP represents a heterogeneous mixture of erythroid and myeloid-primed progenitors, as has recently been demonstrated in single-cell analyses of murine bone marrow (Paul *et al*, 2015; Perié *et al*, 2015).

We also identified distinct groups of cells that were not identified in previous reference datasets. For example, gene expression of C2 was similar to our MEP subgroup, but contained unique upregula-tion of genes whose expression has been previously associated with the development of mast cells (*TPSAB1*, a mast cell specific protease), basophils (*LMO4;* Paul *et al*, 2015), and eosinophils (*MS4A2, FCER1A;* Eon Kuek *et al*, 2016). Notably, in contrast to our GMP-like cluster (C9), C2 cells did not express neutrophil or mono-cytic genes, such as *MPO, ELANE, LYZ*, or *CSF3R* (Lau *et al*, 2005; Rotival *et al*, 2011). We therefore suspected these cells represented Basophil/Eosinophil/Mast cell progenitors (Ba/Eo/Ma), potentially similar to a FCεRIα$^+$ population that was recently identified and validated in human bone marrow as an early precursor of mast cells (Dahlin *et al*, 2016). Cluster 1 was uniquely marked by high expres-sion of clear megakaryocytic (Mk) markers including *PF4, ITGA2B*, and *CLEC1A,* suggesting a putative Mk progenitor identity for C1 cells.

Additionally, our CD34$^+$ subsets also consisted of transitioning populations that are abundant in human hematopoiesis, but lack well-characterized surface markers. For example, C4 cells lacked the expression of mature erythroblast markers, but expressed high levels of *GATA2* and *KIT*, likely representing a transitioning popula-tion along the early stages of erythroid commitment. Similarly, clus-ters 7 and 8 were marked by elevated expression of *FLT3* and *CSF3R* with gradually reduced levels of stem cell markers, likely represent-ing populations similar to the lymphoid-primed multipotent progeni-tor (LMPP) that have previously been described in mouse (Adolfsson *et al*, 2005). These populations also exhibited high expression of L-selectin (*SELL*; *CD62L*), consistent with a recent study which identified CD62L as an early marker of lympho-myeloid fate commitment in human bone marrow (Kohn *et al*, 2012). Taken together, we conclude that our broad sampling strategy enabled us to capture the continuous spectrum of blood differentiation, includ-ing both metastable progenitor states, as well as a spectrum of tran-sitioning cells in between.

### Computational reconstruction of the early hematopoietic fate transitions

While clustering analyses are useful for categorizing cellular hetero-geneity, they impose a discrete framework. We reasoned that the continuity of hematopoietic differentiation (Bendall *et al*, 2011, 2014; Macaulay *et al*, 2016) could enable us to resolve the global topology of the clusters we had identified. We therefore designed a computational strategy for global and unbiased reconstruction of differentiation trajectories from single-cell data (Fig 2A). Our strat-egy was tailored toward the reconstruction of cell-state hierarchies in scRNA-seq data, but shared an approach with pioneering meth-ods that have been developed for recovering branched trajectories in CyTOF data (Qiu *et al*, 2011). Briefly, we began by "micro-clustering" our dataset, subdividing our clusters into bins of 20 cells based on their relative developmental progression. Our choice of $n = 20$ was motivated by the desire to reduce technical noise while maintaining the high resolution of our continuum, and we observed a saturation in micro-cluster similarity with increasing values of $n$. (Materials and Methods; Fig EV2A). These micro-clusters therefore evenly spanned the hematopoietic continuum, yet gene expression variability within an individual micro-cluster was consistent with stochastic Poisson noise generated from a homogeneous population (Fig EV2B–D). Genes that were excluded from clustering analysis also exhibited Poisson noise within a micro-cluster (Fig EV2C). We therefore represent each micro-cluster by the average expression of all its cells, dramatically reducing the noise in scRNA-seq data driven by sparse sampling, with an average of 6,858 genes per micro-cluster.

We then identified the hierarchy of the micro-clusters using minimum spanning trees (MSTs), a commonly used approach for reconstructing topological relationships by learning the most parsimonious set of paths connecting all data points (Qiu *et al*, 2011; Trapnell *et al*, 2014). Notably, we used a bootstrapped approach to assess the reproducibility, by repeatedly running the spanning tree estimation on subsamples of our dataset, and ensuring that our results represent a single robust and highly branched hierarchy. Though the spanning tree is unrooted, we oriented the tree so that the HSC sits atop the hierarchy, while Erythroid (Er), Basophil/Eosinophil/Mast (Ba/Eo/Ma), Neutro-phil/Monocyte (Neu/Mo) and Lymphoid (Lym) progenitors are the terminal leaves (Figs 2B and EV2E). For the hierarchical rela-tionships relating HSC to four downstream fates, we obtained identical results in 100% of the 500 bootstraps. We also obtained similar results using Monocle, which performs dimensionality reduction using independent components analysis, demonstrating that our results were robust across low-dimensional representa-tions of the data (Fig EV2F). However, our bootstrapping revealed inconsistencies in the placement of Mk-committed progenitors (Fig EV2G and H), likely related to their rarity (0.4%), and we therefore focused on the remaining lineages for which we could draw robust conclusions, shown in Fig 2C, along with the dynamic expression levels of illustrative key developmental regu-lators and markers (Fig 2D).

Our computationally reconstructed hierarchy correctly predicts that the initial transition out of multipotency is marked by a reduc-tion in HSC markers, and concomitant upregulation of *CDK6*, which has been shown to control the exit from quiescence in cord blood hematopoiesis (Laurenti *et al*, 2015). We next observed two intermediate states connecting HSC/MPP to the committed progeni-tors. As shown in Fig 2B, the intermediate states did not separate into clear binary fate choices, echoing recent findings that early progenitors exist in a low-primed and continuous state (the "CLOUD"; Velten *et al*, 2017). However, as cells progressed and continued to downregulate HSC markers, we observed segregation

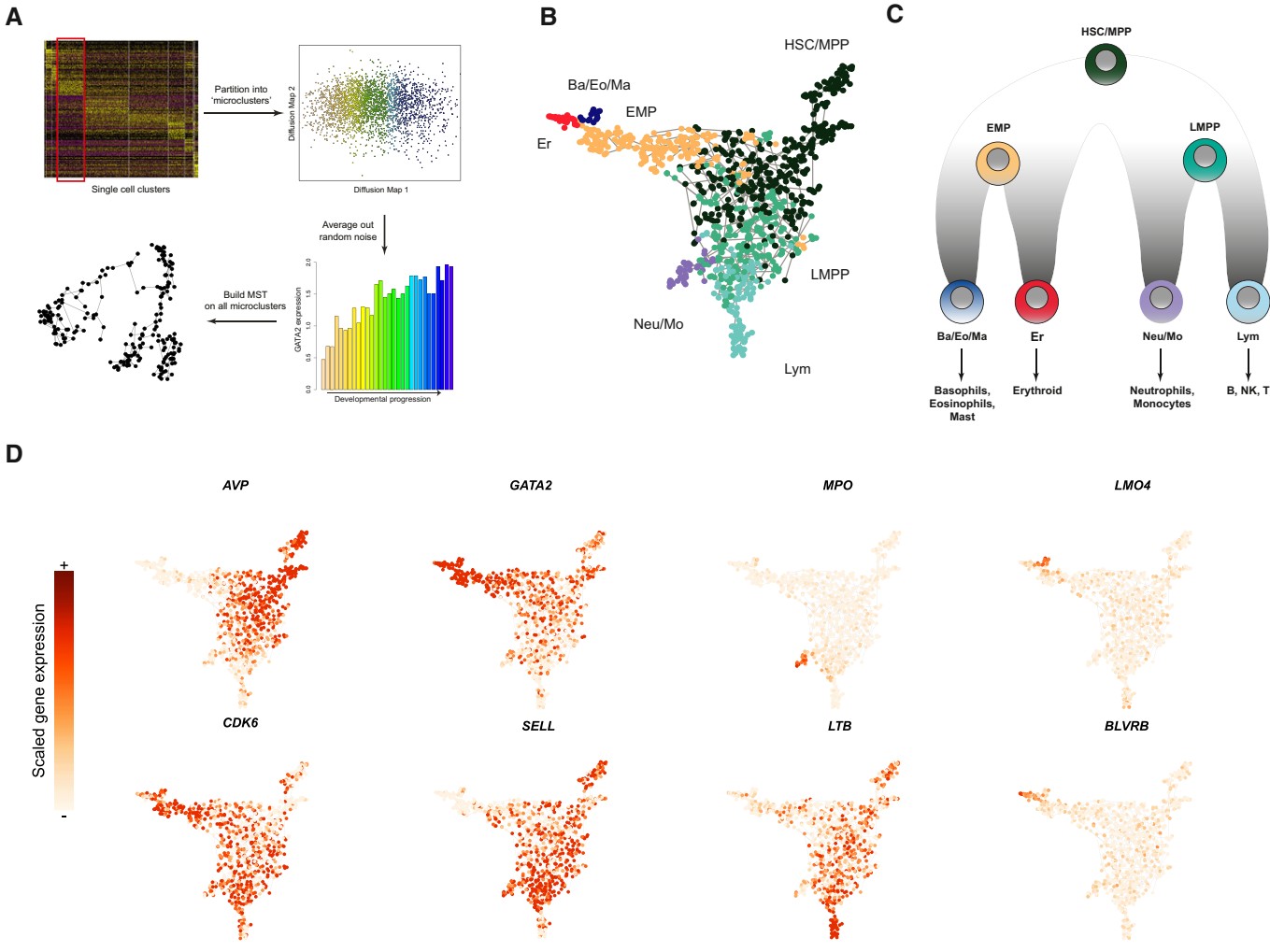

**Figure 2. Reconstructing the topology of early fate decisions.**

A   Schematic workflow for reconstructing cellular hierarchies, consisting of cluster partitioning into "micro-clusters", followed by a minimum spanning tree (MST) procedure built on these micro-clusters, evaluated by bootstrapping afterward.

B   MST computed on 963 micro-clusters, which are colored based on their branch annotation (Materials and Methods), corresponding to the colors in (C).

C   Hierarchical model reconstructed from Drop-seq data (micro-clusters), representing the branching structure in the MST.

D   Expression level of hematopoietic genes overlaid on the cellular hierarchy. These genes include markers of HSC (AVP), differentiating (CDK6) cells, early progenitors (GATA2, SELL), and each of the four downstream lineage progenitors.

of intermediate progenitors into more distinct fate choices (Fig 2B), even prior to the commitment to a single lineage. One path was characterized by the upregulation of transcription factors and cell surface proteins that are commonly associated with the LMPP in both mouse and human, namely, *FLT3, CSF3R,* and *SELL.* This early fate transition is reflected in the differential expression of hundreds of downstream genes (Fig 3A) and is in tight agreement with revised hematopoietic models in both human and mouse that feature a lympho-myeloid-competent progenitor cell lacking erythroid potential (Adolfsson *et al*, 2005; Kohn *et al*, 2012). LMPP cells subsequently subdivided into *MPO*-expressing neutrophil/ monocyte progenitors, or *MME (CD10)*-expressing lymphoid progenitors (Galy *et al*, 1995), recapitulating well-established human hematopoietic models (Doulatov *et al*, 2012; Notta *et al*, 2016).

### Distinct transcriptomic trajectories give rise to distinct myeloid subsets

Alternately, a separate path was marked by the sharp upregulation of *GATA2*, alongside a robust gene set whose expression was widely shared between the Ba/Eo/Ma and Er lineages, but absent from LMPP and Neu/Mo progenitors. These included both well-studied (*GATA1/2, IKZF2, STAT5A*), and putative (*AFF2, ZBTB16, BMP2K, CTNNBL1*) regulators (Figs 3E and EV3A). We therefore conclude that the upregulation of *GATA2* coincides with the entry into an intermediate state resembling an "erythro-myeloid" progenitor (EMP), a recently described oligopotent progenitor state in mouse bone marrow that is largely driven by the activity of GATA factors (Drissen *et al*, 2016). Notably, genes that were induced during the transition from HSC to LMPP—*FLT3, CSF3R, SELL, CD99*—were

absent from the *GATA2*-dependent trajectory, including the Ba/Eo/Ma progenitors, but maintained high levels of expression in Neu/Mo progenitors.

Taken together, our data strongly suggest that different myeloid subpopulations are generated via strikingly distinct trajectories. In particular, the early stages of Neu/Mo differentiation are characterized by molecular features that are shared with early transitions accompanying lymphoid commitment, passing through an LMPP-

like state as previously described in well-established models (Adolfsson *et al*, 2005; Doulatov *et al*, 2010; Kohn *et al*, 2012). By contrast, a smooth path from HSC to Ba/Eo/Ma begins with the induction of *GATA2*, alongside a module of genes that are often associated with the early stages of erythroid development, representing an EMP-like intermediate stage.

While our reconstructed models represent computational hypotheses derived from "snapshot" data, they provide

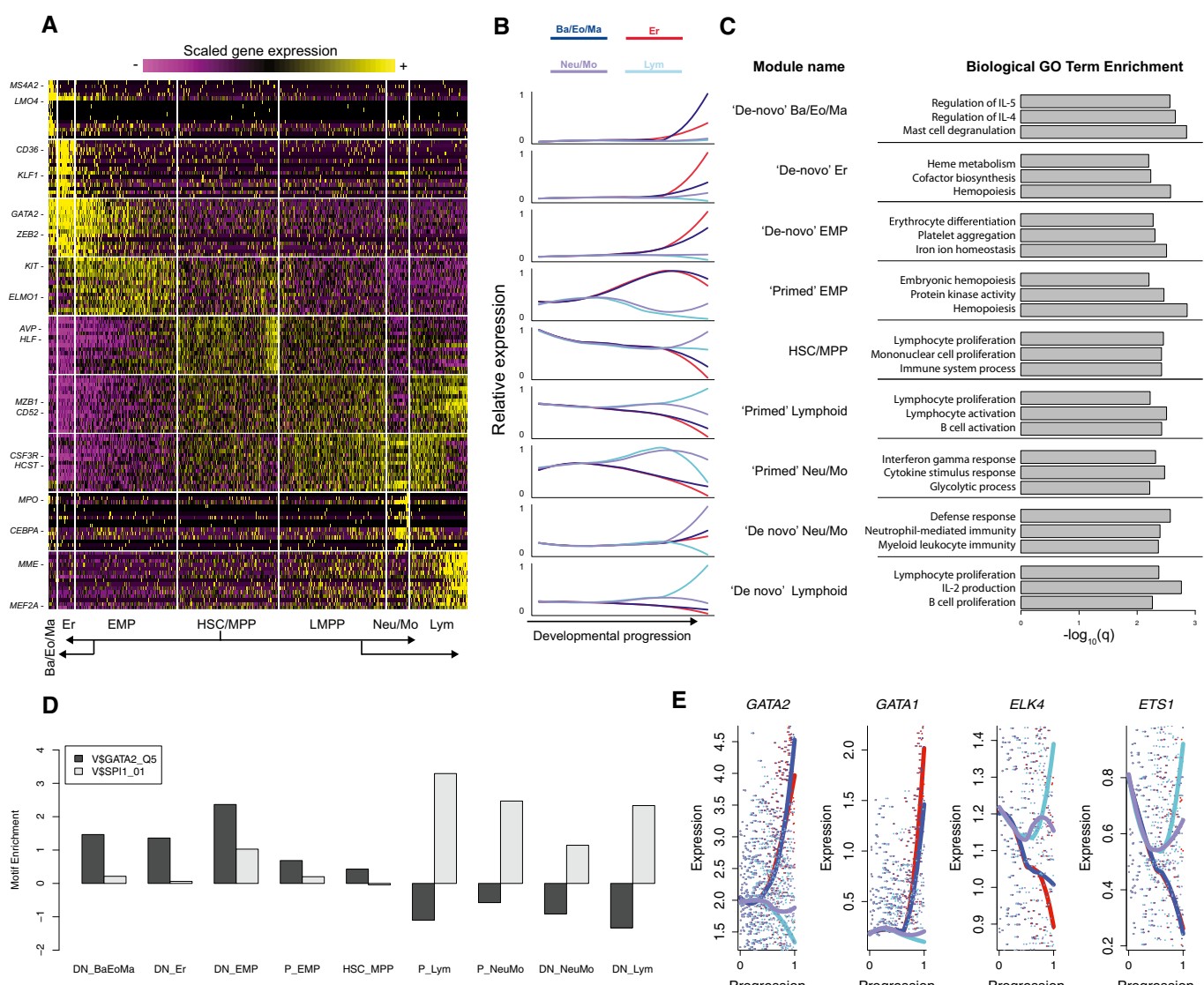

**Figure 3. Transcriptional dynamics during lineage commitment.**

A Heatmap exhibiting transcriptional dynamics of nine distinct gene modules during fate transitions. Data represent scaled gene expression for the 963 micro-clusters, which are grouped by branch annotation, and ordered by increasing developmental progression (distance from the MST root—micro-cluster with the highest *AVP* expression; Materials and Methods). The top 20 markers for each cluster are shown; the module labels for all 517 "branch-dependent" genes are listed in (Table EV3).

B Mean expression for each module, displayed across four trajectories as a function of normalized developmental progression (Materials and Methods), connecting HSC/MPP to each of four downstream lineages.

C Biological GO term enrichments for genes in each transcriptional module.

D We observe enrichments for PU.1 (ETS) or GATA2 motifs in the upstream regions of all nine gene modules. Enrichments correspond to enrichment scores from i-cisTarget (Imrichová *et al*, 2015; Herrmann *et al*, 2012).

E Transcriptional dynamics of GATA and ETS motif binding transcription factors across all four trajectories, using the same color scheme as (B). A local polynomial regression with span = 0.9 was used for smoothing, with the underlying gene expression data shown as individual points. Additional genes are shown in Fig EV3A.

complementary evidence for a recently proposed revised hematopoietic model in mouse, where distinct myeloid subsets are generated through GATA-dependent and GATA-independent pathways. Our results suggest that this model is also relevant for human hematopoiesis (Drissen *et al*, 2016). Indeed, previous functional data also supported a similar model in humans (Görgens *et al*, 2013), but was dependent on surface marker enrichment and is complemented by our unsupervised molecular characterization. As the Ba/Eo/Ma progenitor is missing from standard human hematopoietic models, we therefore intersected markers of this population with the Laurenti reference dataset (Laurenti *et al*, 2013) and observed that our Ba/Eo/Ma markers (e.g., *TPSAB1, HDC*) were most highly expressed in the "MEP" gate, indicating that these progenitors express an immunophenotype that mixes them into a traditional MEP gate (Fig EV3B). As a phenotypic validation, we used flow cytometry to examine the surface phenotype of CD34$^+$ CD117$^+$ FcεRIα$^+$ cells, representing a recently identified mast cell progenitor (Fig EV3C), and found that > 95% of cells in this gate expressed high levels for the transferrin receptor (TFRC/CD71), a marker commonly used to enrich for erythroid-committed cells (Fig EV3D, Dong *et al*, 2011). Taken together, these data suggest that the Ba/Eo/Ma progenitor is transcriptomically and immunophenotypically similar to erythroid-committed cells, consistent with a model where the differentiation trajectories of the two lineages share early upstream molecular transitions.

**Characterizing "primed" and "*de novo*" transcriptional dynamics**

Our reconstructed hierarchy provides a rich scaffold to understand the molecular transitions underlying fate commitment from HSC to downstream lineages. To globally identify genes involved in fate transitions, we designed an unsupervised strategy to select genes with dynamic expression across the hierarchy. Briefly, we assigned pseudo-temporal time points to each micro-cluster according to their positions on the hierarchy, and individually examined the transition from stem cells to each committed lineage to look for genes differentially expressed during individual key fate transition (Materials and Methods). We note that the strategy for gene calling was based on the identification of "branch points" within the topology. This represents a simplification of our data, which show gradual commitment in early progenitors as opposed to distinct and clear branching patterns, but enabled us to perform differential expression. More importantly, when identifying dynamic genes, we placed more weight on micro-clusters that were further from the initial "branch point", consistent with the increasing degree of segregation that we observe in our data. Together, we identified a total of 517 genes having mRNA levels correlated with temporal progression of cell fates. We next performed *k*-means clustering to group together genes with similar transcriptional dynamics across the entire hematopoietic hierarchy, identifying nine gene modules with distinct temporal and lineage patterns (Fig 3A and B; Table EV3).

We first performed GO enrichment analysis (Kuleshov *et al*, 2016) for each module and found that functional enrichments were in broad agreement with expression dynamics, including "hemopoiesis" and "erythrocyte differentiation" for Er-related programs, "defense response" for Neu/Mo modules, and "lymphocyte proliferation" for lymphoid markers (Fig 3C). We also searched the 20-kilobase regions around transcription start sites (TSSs) of genes within each module for over-represented motifs (Herrmann *et al*, 2012; Imrichová *et al*, 2015) and found a strong separation between gene sets whose expression varied between early progenitor states. All gene sets upregulated in the LMPP and downstream cells were strongly enriched for ETS motifs (Fig 3D), which can be bound by PU.1, ETS1, ELK4, and other transcription factor families (Sharrocks, 2001). These motif enrichments mirrored the expression patterns of the transcription factors themselves (Figs 3E and EV3A), as regulators with ETS-binding domains exhibited higher expression in LMPP compared to EMP. In contrast, genes upregulated in the EMP exhibited strong enrichments for GATA motifs, again mirroring the expression dynamics of GATA factors.

Our gene modules also exhibit complex patterns that extend beyond the concept of cluster markers. For example, the final four gene clusters contain genes that are specific to either Neu/Mo or lymphoid progenitors; however, these clusters differ significantly in their expression patterns upstream (Fig 3B). *MME* (CD10) is a canonical marker of lymphoid committed cells (Galy *et al*, 1995). While it is not expressed in early progenitors, it is activated "*de novo*" after the lymphoid restriction ("*de novo*" lymphoid genes). In contrast, the lymphoid-malignancy marker *CD52* (Rodig *et al*, 2006) was detected as early as the HSC and maintained its expression throughout lymphoid differentiation, but was downregulated during fate commitment for all other lineages ("primed lymphoid" genes).

We observed that markers for multiple lineages could also be segregated into "primed" and "*de novo*" programs. In Neu/Mo progenitors, we identified "*de novo*" activation of specific markers and regulators (*MPO, CEBPA*), but also observed "primed" expression of canonical myeloid receptors (*CSF3R, IL17RA*) (Jovanovic *et al*, 1998; Martinez, 2009), whose expression was selectively downregulated during lymphoid commitment. We observed similar patterns for EMP fate commitment as well (Fig 3A and B), allowing us to name other modules as "primed"/"*de novo*" Neu/Mo, "primed"/"*de novo*" EMP, "*de novo*" Er and "*de novo*" Ba/Eo/Ma programs according to their dynamic pseudo-temporal patterns (Table EV3).

Our computational hierarchy suggests that subsets of early progenitors have molecular profiles consistent with transcriptomically intermediate states. Indeed, our LMPP populations co-express the "primed" lineage programs for both lymphoid and Neu/Mo lineages, but have yet to activate the "*de novo*" programs for either lineage. Similarly, the HSC populations co-express primed programs for all downstream lineages simultaneously. Our data are therefore consistent with a model where the early fate transitions in cord blood hematopoiesis pass through molecularly intermediate states, with fate restriction occurring via the downregulation of "primed" expression programs, alongside activation of "*de novo*" genes. Notably, the expression of "primed" genes echoes the functional output of these progenitors populations: HSC can give rise to all downstream lineages, while LMPP lose erythroid potential, concomitant with a downregulation in "primed" EMP genes, in agreement with a model where the presence or absence of multilineage transcriptional priming encodes cellular potential (Hu *et al*, 1997).

## Early lineage priming is conserved between human bone marrow and cord blood

While our developmental hierarchy and transcriptional dynamics were based on CD34$^+$ cells collected from umbilical cord blood, we asked whether our conclusions were unique to this system, or were consistent with CD34$^+$ cells in human bone marrow. Specifically, we wished to ask (i) whether similar transcriptional states and markers could be used to define progenitor populations in both tissues; (ii) if so, whether Neu/Mo and Ba/Eo/Ma myeloid subsets in bone marrow exhibited distinct expression of "primed" programs as we observed in cord blood; and (iii) if present, do "LMPP" cells in bone marrow co-express "primed lymphoid" and "primed Neu/Mo" modules as we see in cord blood. To facilitate this comparison, we applied our recently developed single-cell integration approach to our Drop-seq micro-clusters, and a bone marrow CD34$^+$ dataset profiled with Quartz-Seq (Velten *et al*, 2017). This approach applies canonical correlation analysis (CCA) and non-linear warping techniques to "align" subpopulations that exist in both datasets, based on shared sources of variation (preprint: Butler & Satija, 2017). We have previously demonstrated that this method can successfully align two murine datasets of bone marrow hematopoietic progenitors produced with two different technologies, and therefore asked whether we could detect shared subpopulations between two human hematopoietic tissues.

Indeed, integrated analysis successfully aligned both datasets (Fig 4A), with bone marrow CD34$^+$ cells mapped to each of our Drop-seq progenitor states: HSC/MPP, LMPP, EMP, Er, Ba/Eo/Ma, Neu/Mo, and Lym progenitors (Fig 4B). Strikingly, markers defining these states were extremely well conserved in both datasets, even though they were measured in different tissues, individuals, and technologies (Figs 4C and EV4A), and tSNE visualization was strongly consistent with the same developmental hierarchy as we observed in umbilical cord blood. Cell type proportions were similar between the two tissues, though we did observe a proportional shift toward more committed progenitors in bone marrow, consistent with the results from Notta *et al* (2016) (Fig EV4D). Importantly, we observed that the annotations remained highly consistent when we randomly sampled 500 cells from the bone marrow dataset, and repeated the alignment procedure (Fig EV4E).

We next examined the expression of "primed" programs in the early progenitor subsets. Here, we applied a scoring method developed by Tirosh *et al* (2016) to measure the expression level of a gene module in each cell while controlling for sampling-induced sparsity and dropout. Expression scores for "primed" modules mirrored the patterns we observed in cord blood (Fig EV4B and C). In particular, Ba/Eo/Ma downregulated "primed" genes associated with LMPP, while upregulating primed "EMP" genes compared to HSC/MPP, while Neu/Mo progenitors exhibited opposing patterns (Figs 4D and EV4B). Cells mapping to bone marrow LMPPs also revealed the same multilineage priming phenomenon as seen in cord blood, exhibiting co-expression of "primed" lymphoid and "primed" Neu/Mo modules, and HSC/MPPs were simultaneously enriched for "primed" expression programs for all downstream lineages (Figs 4D, and EV4B and C). Taken together, we conclude that the transcriptional states and molecular transitions we observed in cord blood replicate in an integrated analysis of human bone marrow.

## Epigenetic reinforcement of molecular transitions

While our Drop-seq experiments focus on RNA expression, epigenetic changes, such as the remodeling in chromatin accessibility, are primary determinants of cellular potential. Given the molecular conservation we observed between bone marrow and cord blood, we therefore leveraged a recently published ATAC-seq dataset of human bone marrow hematopoiesis (Corces *et al*, 2016) to integrate chromatin dynamics alongside our transcriptional models. We used ATAC-seq data from HSC, MPP, LMPP, CLP, GMP, and MEP for our analyses, as we have shown (Fig 1C) our Drop-seq clusters are transcriptomically similar to traditional gating for these cell types. Though we did not discover a "CMP" cluster, we included this population in our ATAC-seq analyses as well, as we expect this gate to comprise a heterogeneous mix of EMP and erythroid-committed cells.

We first asked whether the hematopoietic hierarchy calculated from our transcriptomic data was reflected in chromatin accessibility. Each open chromatin region was annotated and linked to the closest TSS, and we identified the 2,000 most variable regions as inputs for principal component analysis (PCA) across progenitor types, which demonstrated that PCs 1 and 2 echoed the structure of our predicted hematopoietic hierarchy (Figs 4E and EV4F). Consistent with this, when we projected ATAC-seq peaks that were adjacent to genes from our transcriptomic clusters onto this PCA, we found that EMP-dependent genes and LMPP-dependent genes projected to opposing sides of the PC1 axis (Fig 4F). Notably, while the ATAC-seq dataset did not contain specific sorting of Ba/Eo/Ma progenitors, genes associated with this lineage showed similar PC1 scores to genes involved in erythroid differentiation, while Neu/Mo-associated genes grouped with lymphoid regulators. Furthermore, when we calculated global gene-level accessibility dynamics in ATAC-Seq (Materials and Methods), we found that these mirrored the transcriptional dynamics we observed for "primed" and "*de novo*" gene modules (Figs 4G and EV4G). ATAC-seq peaks for "primed" genes exhibited high levels of accessibility in early progenitors, with HSC and LMPP exhibiting multilineage epigenetic priming for these loci as well. Accessibility was maintained, however, only during the transition toward a single lineage; for example, we observed sharp decreases in accessibility for primed lymphoid genes in the LMPP to GMP transition, though accessibility was maintained in CLP progenitors. Genes in "*de novo*" modules exhibited low accessibility in upstream progenitors and were specifically remodeled upon transcriptional activation (Fig EV4G). Therefore, intermediate stages exhibit evidence for multilineage priming at both the transcriptomic and epigenetic levels.

While global dynamic patterns were consistent between epigenetic and transcriptomic data, when we closely examined individual gene promoters containing multiple distinct accessibility sites, we observed surprising cases where adjacent peaks had strikingly distinct epigenetic dynamics despite likely regulating the same gene. For example, Fig 4H exhibits how an accessible region adjacent to the *CSF3R* promoter is consistent with its "primed myeloid" transcriptional dynamics, by being enriched in HSC, MPP, LMPP, and GMP. However, another peak 5 kb upstream exhibits opposing patterns and is most accessible in MEP, which is depleted of *CSF3R* transcript according to RNA-seq data. While these "inconsistent" peaks were in the minority (representing 17% of all ATAC-seq

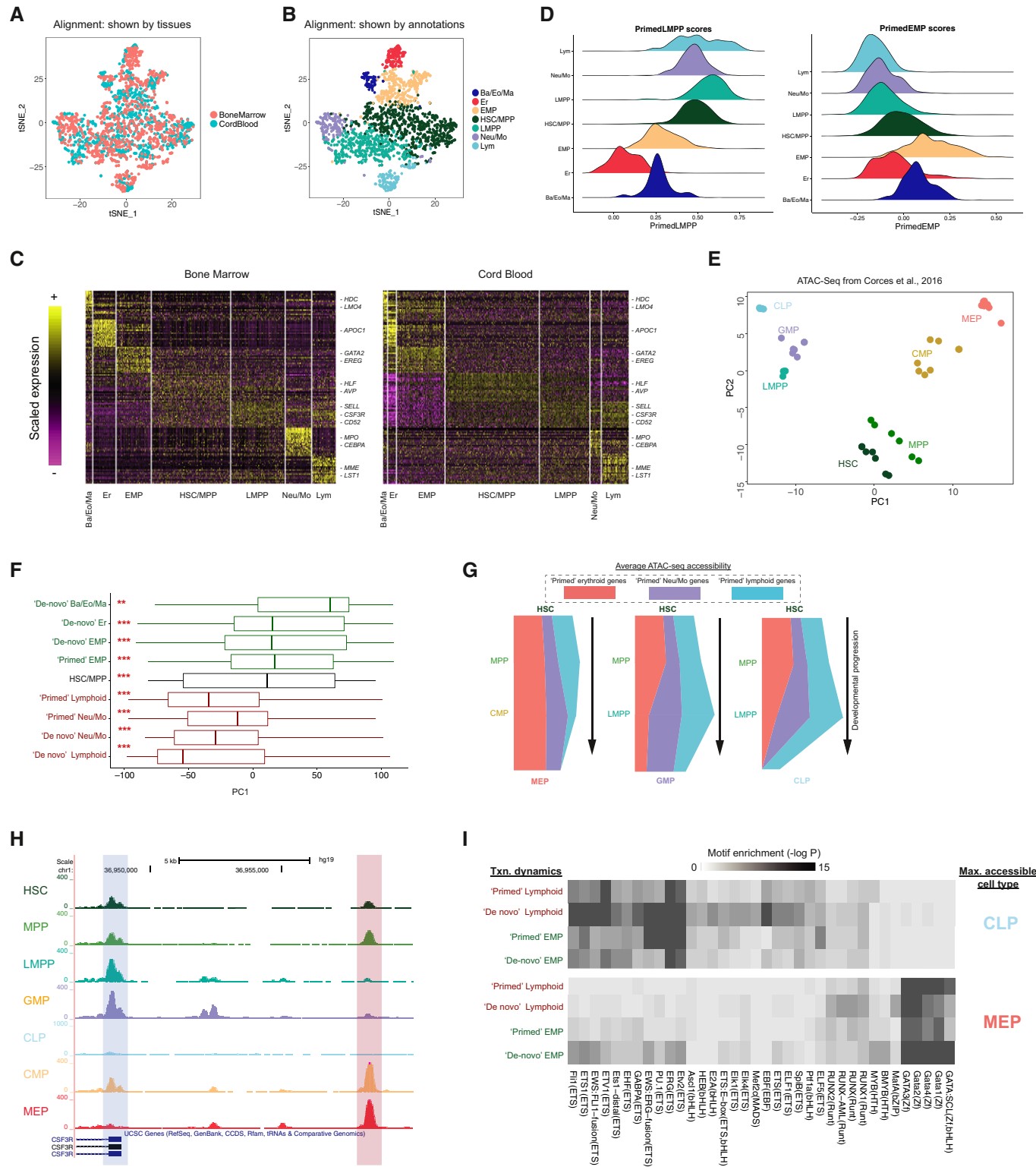

**Figure 4.**

peaks), we observed these patterns for each transcriptomic gene module (Fig EV4H).

To confirm that such inconsistency is global across transcriptomic gene modules, and to explore their patterns of motif

enrichment, we classified all peaks both by the transcriptional patterns of their adjacent gene, as well as by their maximally accessible progenitor type. As an example, we focused on peaks located near lymphoid genes, performing motif enrichment analyses (Heinz

**Figure 4.    Investigating early fate transitions in human bone marrow CD34$^+$ progenitors.**

A    tSNE representation of aligned CD34$^+$ cells from bone marrow, and microclusters from cord blood.

B    Joint annotation of the two integrated datasets (Materials and Methods).

C    Heatmaps showing the expression of top enriched markers shared by annotated progenitors in bone marrow and cord blood. Expression values are scaled (z-scored) for visualization.

D    Ridge plots showing enrichment of expression programs in bone marrow CD34$^+$ progenitors. Cells are colored and grouped by annotated progenitor types, and relative enrichment is represented by a scoring method from Tirosh et al (2016). Left: enrichment for "primed" LMPP genes ("primed" Lym and "primed" Neu/Mo genes); Right: "primed" EMP genes.

E    PCA of ATAC-seq data from Corces et al (2016). Multiple points per cell type correspond to biological replicates. Variation along PC1 echoes the first fate bifurcation in Fig 2B.

F    Projections of nine transcriptomic gene modules onto ATAC-seq PCA in (E). Modules segregate into two groups, with either significantly positive or negative PC1 scores, that are consistent with transcriptional dynamics in Fig 3A. Asterisks indicate that gene scores are significantly different from zero (***$P < 10^{-5}$, **$P < 0.01$; Kolmogorov–Smirnov test). Vertical lines (left to right): first quartile, median, third quartile; whiskers: data points outside the first and the third quartiles.

G    "River" plots, exhibiting quantitative remodeling of chromatin accessibility during differentiation from HSC into three downstream lineages. Width of the river corresponds to the average accessibility for "primary" peaks in this module (Materials and Methods). Peaks adjacent to "primed" genes are accessible for all lineages in early progenitors, but are maintained in only a single lineage during differentiation.

H    Screenshot from UCSC genome browser showing data from Corces et al (2016). A peak near the TSS (blue shading) shows dynamics consistent with the "primed Neu/Mo" transcription of CSF3R, while another peak upstream (red shading) shows the opposing ("inconsistent") dynamics.

I    HOMER motif enrichments for accessible regions, grouped by both transcriptional dynamics (Fig 3A) and maximally accessible cell type (CLP or MEP from Corces et al, 2016). GATA motifs are heavily enriched in MEP-accessible peaks, even if located adjacent to lymphoid-specific genes.

et al, 2010; Materials and Methods) on peaks that were maximally accessible in the CLP ("consistent peaks"), and peaks that were maximally accessible in MEP ("inconsistent" peaks). We found that consistent peaks were strongly enriched for ETS and bHLH motifs, though we also discovered specific enrichments for lymphoid regulators (i.e., EBF1; Fig 4I). Inconsistent peaks, however, lacked ETS motifs and exhibited striking enrichment for GATA motifs instead, even though they were adjacent to the same set of genes. When reversing the analysis and focusing on EMP genes we observed the same phenomenon: "Inconsistent" peaks lacked GATA motif enrichment, and instead contained motifs for lymphoid lineage regulators. These analyses suggest that the accessibility of these peaks is driven primarily by the motifs they contain, instead of the transcriptional dynamics of the adjacent gene. The minority of "inconsistent" peaks may not affect gene expression, or alternately, they may suggest cross-antagonism in early fate transitions by potentially acting as repressive elements. ETS and GATA-binding factors have been previously demonstrated to act as both activators and repressors for a subset of key promoters (Starck et al, 2003), and future functional experiments will illuminate if a subset of these "inconsistent" peaks represent similar phenomenon.

## Coupling single-cell immunophenotyping with transcriptomics

We wondered how cells in our Drop-seq clusters might fall into traditional gating strategies, and also whether our Drop-seq data could be used to propose novel surface markers to enrich for early progenitors primed to different fates. We therefore designed a strategy to couple the unbiased nature of our Drop-seq data with traditional immunophenotyping assays, as shown in Fig 5A. Single cells were isolated into individual wells of 96-well plates using index-sorted FACS, enabling us to measure and store the immunophenotype of each cell. We next processed single cells using a modified version of the SMART-Seq2 protocol (Materials and Methods). In order to map the transcriptomes of these cells to our Drop-seq progenitor populations, we used a random forest-based classifier (Wright & Ziegler, 2015) to assign each cell onto one of our hematopoietic stages from the Drop-seq data (Fig 2B). Effectively, this strategy enabled us to project the index sorting data onto our hematopoietic hierarchy.

We first sorted a plate of CD34$^+$, CD117$^+$, FcεRIα$^+$ cells (Fig EV3C), representing a recently identified mast cell specific progenitor, and indexed on the level of TFRC (CD71), a surface protein commonly used as marker for Er-committed cells (Dong et al, 2011). We saw that 83% of these cells projected to our Ba/Eo/Ma progenitor (Fig 5B), and the group overall exhibited high expression of "de novo" EMP and "de novo" Ba/Eo/Ma transcriptional programs. Alongside our results showing CD71 expression on mast cell progenitors (Fig EV3D), we therefore conclude that our Ba/Eo/Ma population does represent a "bona fide" granulocyte progenitors, and has traditionally fallen within "MEP" standard gates.

We then focused our experiments on CD34$^+$ CD38$^-$ CD45RA$^+$ cells (expected to represent LMPP and early downstream progenitors) and selected 865 profiled cells after filtration, together with the indexed protein levels of the well-characterized lymphoid marker CD10, as well as two putative markers from our Drop-seq data, CSF3R and CD52. These putative markers derive from the "primed" lymphoid and "primed" Neu/Mo programs, respectively, and have not, to our knowledge, been previously used to subdivide early CD38$^-$ hematopoietic progenitors.

After mapping our plate-based scRNA-seq data to the Drop-seq hierarchy, as expected, we observed that CD34$^+$ CD38$^-$ CD45RA$^+$ cells strongly enriched for LMPP, Neu/Mo, lymphoid, and HSC/MPP groups, with negligible mapping to EMP or downstream precursors. We observed an even stronger enrichment of lymphoid progenitors when examining the subset of cells that stained positive for CD10 (Fig 5B), validating its suitability as a marker for lymphoid commitment, as well as our coupling and projection strategy. However, we observed that CD10$^-$ cells remained transcriptionally heterogeneous, and projected to uncommitted, myeloid (Neu/Mo), and lymphoid fates at roughly equal proportions. Supporting these conclusions, an independent principal component analysis (PCA) using the only plate-based scRNA-seq data of LMPP separated correlated gene sets associated with myeloid and lymphoid commitment (Fig EV5B).

To explore whether we could further subdivide the LMPP gate based on novel markers, we examined the distribution of CD52 and CSF3R protein expression for cells projecting to distinct transcriptomics clusters (Fig 5C; additional markers in Fig EV5A). In agreement with our predictions from the Drop-seq data, CD52 protein

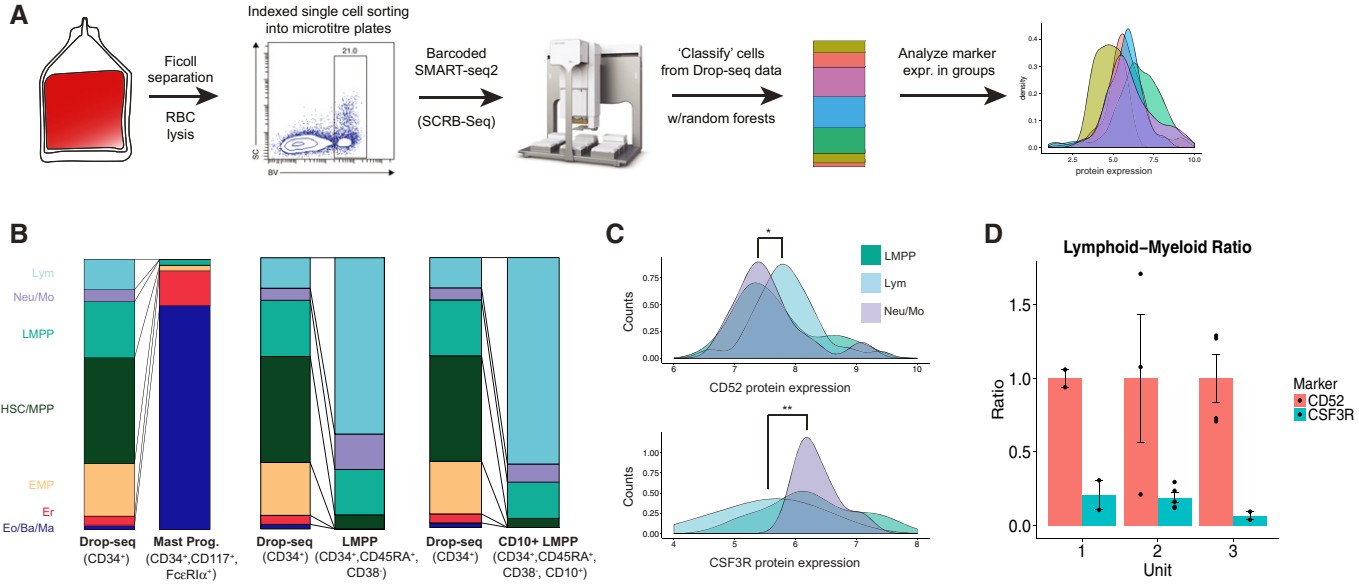

**Figure 5. Projecting cellular immunophenotypes onto Drop-seq data.**

A  Schematic to generate scRNA-seq data using index-enabled FACS sorting into 96-well plates, and to project this data onto the Drop-seq hierarchy to evaluate surface marker expression.

B  Compositional makeup of SMART-Seq2 experiments, after projection onto Drop-seq data (Materials and Methods). Height of each bar segment indicates the percentage of cells that map to each Drop-seq annotation. To facilitate visual comparison, the background distribution of all CD34$^+$ cells is shown three times.

C  Distribution of indexed protein levels for CD52 (top) and CSF3R (bottom) for cells mapping to three Drop-seq branches. Protein expression is shown in log-scale. $**P < 10^{-3}$, $*P < 0.05$ (Kolmogorov–Smirnov test).

D  We observe a significant ($P < 10^{-3}$; Welch two-sample *t*-test) relative depletion of lymphoid progeny (CD19$^+$ B cells and CD56$^+$ NK cells) from CD52$^-$ CSF3R$^+$ progenitors, compared to CD52$^+$ CSF3R$^-$ progenitors after *in vitro* differentiation with the MS5-MBN assay. Barplot shows results from three separate cord blood units (error bars reflect SE based on all replicates within each unit, which range from two to five depending on total cell number).

expression was significantly ($P < 0.05$; Kolmogorov–Smirnov test) higher in lymphoid-primed cells compared to myeloid-primed cells, while CSF3R protein levels exhibited the opposite pattern ($P < 1e^{-3}$; two-sided Kolmogorov–Smirnov test). While significant, no protein marker was in perfect agreement with our clustering, and it is unlikely that CSF3R alone or CD52 could be used to sort transcriptomically pure populations, echoing recent studies in mouse (Paul *et al*, 2015; Nestorowa *et al*, 2016), demonstrating the challenge to define FACS panels that can perfectly represent the complexity of a single-cell transcriptomics readout.

Given that CD52 and CSF3R levels correlate with the transcriptomic profile of LMPPs, we asked whether these could potentially serve as new surface markers to distinguish the functional output of these early progenitors. To test this, we sorted populations of CD34$^+$ CD38$^-$ CD45RA$^+$ CD10$^-$ LMPPs, after gating on CD52 and CSF3R, into individual wells and performed *in vitro* differentiation using the MS5-MBN assay (three cord blood units from different individuals, two to five biological replicates per unit). After 3 weeks, we compared the output of CD52$^+$ CSF3R$^-$ and CD52$^-$ CSF3R$^+$ LMPPs using flow cytometry, analyzing CD19, CD56, and CD11b (B/NK/myeloid cells) to assess lymphoid and myeloid outputs (Fig EV6A and B, Materials and Methods, Laurenti *et al*, 2013). While we observed heterogeneity in the lymphoid/myeloid ratios for individual units, in each case, CD52$^-$ CSF3R$^+$ progenitors gave rise to significantly fewer lymphoid cells, in agreement with our transcriptomic predictions (Figs 5D and EV6). We did not observe exclusive lymphoid or myeloid output from any replicates,

demonstrating that CSF3R or CD52 do not perfectly enrich for pure subpopulations. However, these data validate that the expression of primed programs correlates with the functional potential of early progenitor populations.

## Discussion

In this study, we applied massively parallel single-cell sequencing to dissect cellular heterogeneity in CD34$^+$ progenitors in human cord blood. We leverage our dataset to reconstruct molecular trajectories to four downstream lineages. We identify striking heterogeneity in the early molecular transitions toward the commitment of distinct myeloid cell subsets and observe distinct transcriptional dynamics for "primed" gene modules that are conserved in human bone marrow, echoed in chromatin accessibility, and correlate with functional outputs as measured through *in vitro* differentiation assays.

Taken together, our results do show substantial agreement with key tenets of the pioneering and canonical models of hematopoiesis, albeit with a slightly refined structure. Indeed, our findings do imply that cells undergo lineage selection in a gradual way, passing through at least a small set of intermediate states whose transcriptomic and open chromatin makeup promotes plasticity. We do not, however, find evidence of a "CMP" state, and our results here more closely match with murine scRNA-seq and functional studies (Paul *et al*, 2015; Drissen *et al*, 2016). In fact, the agreement between these datasets suggest that human and murine hematopoiesis may

be more similar than once thought, though this conservation was difficult to observe using traditional surface markers, which vary widely between species (Wu *et al*, 2014).

Our results suggest that key portions of the transcriptional networks driving fate commitment are conserved between cord blood and bone marrow hematopoiesis, and we expect that key regulators play similar roles in both tissues. For example, a recent study identified a genetic variant in GATA2 that affected the basophil and eosinophil, but not neutrophil and monocyte, cell counts in an adult human GWAS (Guo *et al*, 2017). However, our data support the idea that the density of cells (early progenitors vs. later unipotent precursors) is likely altered between the tissues, explaining the greater percent of oligopotent cells observed in cord blood (Notta *et al*, 2016). Additionally, the reproducibility we observe in the density across cord blood units suggests the attractive nature of this system for reproducible analyses, as even bone marrow samples within an individual will inevitably vary in composition based on niche-dependent sampling.

Finally, our data reveal a transcriptomic map of fate decisions taken by early progenitors. However, these sequencing data alone cannot reveal the fundamental mechanism by which each single cell initiates a decision. Recent work has convincingly demonstrated that the early hematopoietic decisions cannot be completely explained by PU.1/GATA1 ratios in early progenitors (Hoppe *et al*, 2016). More likely, broader groups of transcriptional regulators work in concert to initiate and establish lineage selection, with additional key inputs coming from inherited chromatin state and environmental signals. Single-cell technologies continue to develop, with exciting advances that pair sequencing with genetic perturbations (Dixit *et al*, 2016), lineage barcodes (Wu *et al*, 2014), and spatial information (Chen *et al*, 2015). We therefore anticipate that future studies will derive a deep and integrated understanding of the role of gene expression, epigenetic state, lineage, and environment on cell fate decisions.

# Materials and Methods

### Cord blood processing

Umbilical cord blood from anonymous healthy donors was obtained through the National Cord Blood Program from New York Blood Center. Within 48 h after cord blood collection, mononuclear cells (MNCs) were isolated from each cord blood unit by density centrifugation using Ficoll-Paque PREMIUM (GE Healthcare #17-5442-03), and enriched for CD34$^+$ cells using MACS separation (Miltenyi Biotec #130-100-453). Briefly, umbilical cord blood was diluted twofold using DPBS without calcium and magnesium (Corning #20-031-CV), layered on top of 15 ml Ficoll-Paque PREMIUM in a 50-ml Falcon tube, and spun down at 850 *g* for 30 min at room temperature with the brake off. The mononuclear cell layer was then isolated and washed with MACS buffer (DPBS with 0.5% BSA, Sigma-Aldrich #A8806-5G) after red blood cell lysing with ACK lysing buffer (Life Technologies #A10492-01). MNCs were then enriched for CD34$^+$ cells by incubating them with magnetic beads conjugated to mouse anti-human CD34 antibody for 30 min, and passed through a magnetic MACS LS column (Miltenyi Biotec #130-042-401). CD34$^+$ cells were bound to the LS column and later

flushed off and collected in 5 ml MACS buffer. For Drop-seq experiments, two consecutive enrichment steps were performed to increase the purity of enriched CD34$^+$ cells.

### Cell preparation and scRNA-seq

For Drop-seq, enriched CD34$^+$ cells after MACS separation were diluted to 200 cells/μl in PBS-0.1% BSA solution in single-cell suspensions, and loaded into the Drop-seq device (Macosko *et al*, 2015). We set the input concentration of cells and beads to be 200 cells/μl and 250 beads/μl, and optimized the flow rates for cells (2,500 μl/h), beads (2,500 μl/h), and oil (6,800 μl/h) to obtain a stabilized aqueous flow. We also ensured that these flow rates returned a low cell doublet rate (1–2%) using human/mouse species-mixing experiments as suggested, using human HEK293 cells and mouse 3T3 cells. Droplets were collected after each run, and we recovered single-cell transcriptomes attached to microparticles (STAMPs) using 6× SSC and perfluorooctanol (PFO, Sigma #370533). Reverse transcription was performed on the STAMPs in a pooled fashion using Maxima H Minus Reverse Transcriptase (Thermo Fisher Scientific #EP0752), followed by exonuclease I cleavage to remove primers not bound to mRNAs. cDNAs were then amplified through PCR using KAPA HiFi HotStart ReadyMix (Kapa Biosystems #KK2602) by collecting 5,000 STAMPs per PCR reaction, and later fragmented and prepared into paired-end sequencing libraries with the Nextera XT DNA sample prep kit (Illumina) using custom Read 1 primers (GCCTGTCCGCGGAAGCAGTGGTATCAA CGCAGAGTAC, IDT). Libraries were quantified using Qubit and BioAnalyzer High Sensitivity Chip (Agilent) and sequenced on the Illumina HiSeq 2500 machine.

For index sorting experiments with LMPP, CD34$^+$ cells were enriched from umbilical cord blood using MACS separation and stained with the following antibodies in single-cell suspensions: APC mouse anti-human CD34 (BD #560940, clone 581), Alexa Fluor 700 mouse anti-human CD38 (BD #560676, clone HIT2), APC/Cy7 anti-human CD45RA antibody (BioLegend #304127, clone HI100), CD52 monoclonal antibody FITC (Thermo Fisher Scientific #MA1-82037, clone HI186), BV421 mouse anti-human CD10 (BD #562902, clone HI10a), and PE mouse anti-human CD114 (CSF3R, BD #554538, clone LMM741). The amount to use per antibody was determined from titration experiments using cord blood MNCs or enriched CD34$^+$ cells. CD34$^+$ CD38$^-$ CD45RA$^+$ cells were gated on the SONY SH800Z cell sorter with CD10, CD52, and CSF3R indices recorded, and individual cells were sorted into single wells in 96-well plates. For mast cell progenitors, MNCs in single-cell suspensions were stained with the following antibodies: APC mouse anti-human CD34 (BD #560940, clone 581), APC/Cy7 anti-human CD45RA antibody (BioLegend #304127, clone HI100), BV421 mouse anti-human CD117 (BD #562435, clone YB5.B8), PE anti-human FcεRIα antibody (BioLegend #334609, clone AER-37), and FITC Mouse Anti-Human CD71 (BD #561939, clone M-A712). We gated CD34$^+$ CD117$^+$ FcεRIα$^+$ CD45RA$^-$ and CD34$^+$ CD117$^+$ FcεRIα$^+$ CD45RA$^+$ cells on the SONY SH800Z sorter, with CD71 index recorded, and single cells were isolated into individual wells on 96-well plates.

Cells were immediately lysed and mRNAs were released when single cells were sorted into wells with 5× Maxima reverse transcription buffer, dNTP mixture, RNase inhibitors (SUPERase In RNase Inhibitor, Thermo Fisher Scientific #AM2696), RT primers

and water. We reverse-transcribed the mRNAs using Superscript II Reverse Transcriptase (Thermo Fisher Scientific #18064071), and amplified cDNAs for each cell (KAPA) in individual wells using the SMART-Seq2 protocol (Picelli *et al*, 2013), with the exception that a 12-base cell barcode was included in the 3′-end RT primer. This allowed us to perform multiplexed pooling before library preparation with the Nextera XT DNA sample prep kit (Illumina), and returned 3′ biased data similar to the Drop-seq protocol, enabling direct comparison. We quantified the cDNA libraries on Agilent BioAnalyzer and sequenced them on HighSeq 2500 with paired-end sequencing.

## Raw data processing and quality control

Raw reads from scRNA-seq were processed using Drop-seq tools v1.0 (Macosko *et al*, 2015). Briefly, reads were mapped to the human hg19 reference genome, and a digital expression matrix was returned with counts of unique molecular identifiers (UMIs) for every detected gene (row) per cell barcode (column). To determine the number of cells (cell barcodes) represented in the expression matrix, we used the elbow plot method recommended by the Drop-seq core computational protocol, which utilize the cumulative distribution of reads and identify an inflection point in the plot. Beyond the inflection point should only be empty micro-particles exposed to ambient RNA.

We further filtered cells by removing those with less than 1,000 UMIs detected, and those with transcriptomic alignment rates less than 50%. We also calculated the percentage of reads aligned to mitochondrial genes per cells and removed cells with greater than 10% of UMIs corresponding to mitochondrial genes (Ilicic *et al*, 2016). A higher proportion of mitochondrial genes indicates the loss of cytoplasmic mRNAs and/or high cell stress experienced during sample preparation (Ilicic *et al*, 2016). The filtered matrix was then log-normalized to correct for the difference in sequencing depth between single cells, by applying the formula below to each cell barcode, where $c_i$ indicates the raw counts for gene $i$:

$$\text{Normalized expression} = \ln\left[\left(\frac{c_i}{\sum_i c_i} + 1\right) \times 10{,}000\right]$$

Variation in cell cycle stages can contribute to the heterogeneity in single-cell data and will be confounded with developmental heterogeneity. Furthermore, technical factors will also act as confounding sources of noise when analyzing heterogeneous populations. We therefore sought to remove cell cycle effects together with technical covariates through latent variable regression (Buettner *et al*, 2015). Briefly, we assigned a cell cycle score for every cell from a principal component analysis (PCA) done using only a published list of genes whose expression level is strongly correlated with cell cycle phase (Macosko *et al*, 2015), from which we found that both PC1 and PC2 represented the separation between S and G2/M phases. We then modeled the expression for each gene $i$ using the formula:

$$G_i = \beta_0 + \sum_j \beta_j X_j + \varepsilon_i$$

Where $G_i$ is a vector showing the log-normalized expression for gene $i$ in all the cells, $X_i$ represents a user-defined covariate to regress out and $\varepsilon_i$ is the random noise associated with gene $i$. In addition to cell cycle scores, we have chosen to include total UMI counts, alignment rates, percentage for mitochondrial reads, and donor IDs as input variables for regression. The residuals were then *z*-scored and used as corrected expression values for dimensionality reduction, which is described below.

## Dimensionality reduction

From the normalized expression matrix, we first identified a set of variable genes with high dispersion rates across cells. Briefly, we calculated the mean per gene in the non-log space, and dispersion was calculated from dividing mean by variance. We selected 5,000 genes with the highest dispersions as variable genes for dimensionality reduction, a common step in single-cell data analysis for reducing noise and capturing biological signals. Here, we leveraged independent component analysis (ICA), which was initially developed to separate a group of mixed signals into additive sources that are independent of each other, and has more recently been applied to dimensionality reduction for single-cell data (Trapnell *et al*, 2014). We implemented ICA using the ica package in R.

The returned ICs contain pooled information across multiple correlated genes and thus represented "meta-genes" (Setty *et al*, 2016), which were robust to dropout events in scRNA-seq data. We noticed that the variance accounted for by each component fell after IC25. Furthermore, genes with strong IC8 loadings were dominated by mitochondrial genes, and we therefore used ICs 1–25 (excluding IC8) for downstream analysis.

## Clustering of single cells

The CD34$^+$ population contains hematopoietic stem and progenitor cells which are expected to be transcriptionally heterogeneous (Broxmeyer *et al*, 1989; Gluckman *et al*, 1989; Nimgaonkar *et al*, 1995), and therefore, we used clustering analysis to reveal the different transcriptomic states within the cord blood CD34$^+$ pool. We utilized biologically relevant ICs from dimensionality reduction as input for clustering, which we achieved by leveraging a modularity-based method on shared nearest-neighbor (SNN) graphs (Blondel *et al*, 2008; Xu & Su, 2015). We defined the similarity of cells based on the overlap of neighborhoods (proportion of shared neighbors), which were built on Euclidean distances from the 24 input ICs/meta-genes. An SNN graph was then constructed using Jaccard similarity. In this SNN graph, groups of cells with largely overlapping neighborhoods represent interconnected "communities" in a network, and therefore exhibit similar transcriptional patterns (Levine *et al*, 2015; Xu & Su, 2015). To partition the graph into a set of clusters, we utilized modularity optimization to find the best assignment for each cell through multiple iterations, where modularity (Q, shown below) evaluates both inter-cluster- and intra-cluster connectivity on a graph (Blondel *et al*, 2008).

$$Q = \frac{1}{2m} \sum_{i,j} \left[A_{ij} - \frac{k_i k_j}{2m}\right] \delta(c_i, c_j)$$

Specifically, $A_{ij}$ refers to the edge weight between nodes $i$ and $j$, $k_i$ is the sum of all edges to node $i$ ($k_i = \sum_j A_{ij}$), $m = \frac{1}{2}\sum_{ij} A_{ij}$, $\delta(c_i, c_j) = 1$ if $c_i = c_j$ (both cell $i$ and $j$ are assigned to the same cluster) and 0 if otherwise. By setting $k$ (the number of nearestneighbor to define a neighborhood) = 25, resolution = 1.0 (which determines the number of clusters being returned) and 100 random starts, we obtained 21 single-cell clusters using the function FindClusters() in Seurat package, implemented from a previously published modularity optimizing software (Waltman & van Eck, 2013).

We note this clustering imposes a discrete framework on the data. While a set of clusters can be useful for interpretation of single-cell data, our using of clustering algorithms does not preclude the potential for the underlying data to fall along a continuous manifold. Indeed, in downstream analyses, we further subdivide the clusters to better represent a more continuous landscape of cellular differentiation. However, we find this clustering framework to be valuable for interpreting and evaluating our data, specifically, to compare to previously generated microarray datasets, and to compare cellular densities across different cord blood units (Fig 1C and D). Additionally, this clustering enables us to remove rare contaminant populations of differentiated cells that passed through the CD34 column. For example, cells in cluster 16 were highly expressing T cell genes such as *CD6, CD3D, CD247,* and *CD2*. Cluster 11 was enriched in genes unique for B cells— *MS4A1, CD83, CD22,* and *CD79A*, while lacking *MME (CD10)* expression, indicating that cells in cluster 11 were committed to B-cell differentiation (Fig EV1B). Overall, we kept 12 clusters that represented early progenitors of megakaryocyte, erythrocyte, lymphoid, and myeloid cells (19,394 from 21,306 cells) for downstream analysis. In two cases, we observed that two clusters shared the same set of markers but differed primarily in the quantitative levels of these, and we therefore merged these two pairs of clusters together to result in a final set of 10 clusters for downstream analysis.

To ensure that our clustering results represented the structure in our data as opposed to exact parameter values, we performed a robustness test to assess whether pairs of cells that clustered together in the original analysis also clustered together if we modified parameter values. We therefore ran 25 clusterings on the dataset, over the combinations of five resolution/granularity values (0.8, 0.9, 1, 1.1, 1.2) and five values for the number of neighbors in the initial graph ($k$ = 15, 20, 25, 30, 35). Visualizing these results in Fig EV1C, we observe that cell pairs that clustered together in the original analysis consistently clustered together across analyses, particularly for more committed populations where the boundaries between cell states are more clear.

### Evaluation of cluster identities

We next sought to compare the gene expression patterns of our single-cell clusters with previously characterized progenitor populations in human cord blood. We used a previously published microarray reference dataset (Laurenti *et al*, 2013), which contains expression profiles of CMP, megakaryocyte-erythroid progenitor (MEP), HSC, granulocyte–monocyte progenitor (GMP), and MLP. We hypothesized that if of our single-cell clusters matched any of these reference populations, the two groups should share common markers of gene expression. We reasoned that the most informative markers would represent genes that were not only upregulated in expression for a given cell group, but would in fact be most highly expressed in this group compared to all other groups.

We therefore leveraged the published list of gene expression signatures for the dataset, extracting the top 250 genes that were most significantly upregulated in each population (as originally computed from limma (Ritchie *et al*, 2015). To define markers for each reference subpopulation, we required that the gene not only be in this upregulated list, but also be expressed at the highest level across the dataset.

After defining these markers of reference populations, we examined the expression of these genes in our single-cell clusters, identifying which single-cell cluster had the highest expression for most of these markers. For example, of the 143 "reference markers" for GMP from the microarray dataset, 74 of these were most highly expressed in cluster 9 cells ($P < 10^{-45}$; one-sided test of equal proportions). Figure 1C shows the results of this analysis for all pairs of single-cell and reference clusters. With this method, we could recover well-characterized progenitor states from our Drop-seq clusters using the reference dataset.

### Determining unbiased marker sets

In the previous analysis, we examined the expression of previously identified markers in our single-cell clusters. Alternatively, we can also identify markers that define our single-cell states unbiasedly. We defined an unbiased set of markers using a likelihood ratio test that is specifically designed for zero-inflated data (McDavid *et al*, 2013) and that we have previously applied to Drop-seq (Macosko *et al*, 2015). This test was run on the "normalized" expression data, and we present this list of markers in Table EV1. As a non-parametric alternative, we can also identify genes that are upregulated in each cluster based on the "corrected" expression levels (after latent variable regression). Here, we average the scaled residuals after regression for all genes within each cluster and select the genes with the highest average score as cluster markers, after removing ribosomal and mitochondrial genes. Though this is not based on a statistical test, we found that these marker sets were more informative, as they were performed on the corrected data. We report the top 100 markers for each cluster in a separate tab on Table EV1.

### Micro-clustering

Our Drop-seq dataset should sample both cells in metastable progenitor states, as well as cells which are transiently progressing through a differentiation hierarchy. Indeed, this logic suggests that we can reconstruct developmental histories from cellular snapshots of many single cells. This has been the underlying logic for many trajectory building algorithms, such as Monocle, Wanderlust, and Wishbone (Bendall *et al*, 2011; Trapnell *et al*, 2014; Setty *et al*, 2016; Qiu *et al*, 2017). Importantly, the assumptions underlying this strategy require that we sufficiently sample the process to capture both abundant and rare transition states and that our sampling procedure does not exclude particular states based on prior enrichment. The scale of our Drop-seq

datasets, combined with the relatively unbiased strategy for sample preparation, strongly supports these assumptions for our analyses.

While our clustering analyses are valuable for interpreting the major transcriptional states in a complex system, they impose a discrete framework on a transitioning cellular population. Moreover, the precise number of clusters for any algorithm is dependent on the granularity parameters used. We therefore reasoned that even within the clusters we defined in Fig 1, we should observe developmental heterogeneity, with each cluster consisting of both "early" and "late" cells.

To address this, we developed a strategy to "micro-cluster" our data, further subdividing our clusters into small groups of cells that not only mapped to the same cluster identity, but also were in a similar developmental state. Therefore, within each cluster, we ran a diffusion map procedure (Coifman & Lafon, 2006) using the Euclidean distance defined by all mRNA markers' expression. For each cluster, we found that the eigenvalues dropped off quickly after the first two diffusion map components (DMCs) within a cluster, and exhibited a unidirectional path, consistent with developmental heterogeneity. We fit a principal curve on DMCs 1 and 2 using the principal.curve() function in the R princurve package with default parameters (Hastie & Stuetzle, 1989). The progression of each cell was defined by projecting cells onto the principal curve, and we separate a cluster into small groups of 20 cells ordered by principal curve projection using the cut2() function in R. In this way, we partitioned our original dataset into 963 "micro-clusters". We took the mean of the normalized expression for all detected genes, forming a new expression matrix of 30,730 genes and 963 micro-clusters, dramatically reducing the sampling noise associated with single-cell data. We sought to select the number of cells per micro-cluster ($n$) at a level which reduced the Poisson noise without blurring distinctions in the dataset. To guide this selection, we computed the maximal correlation and covariance between pairs of micro-clusters as a function of different $n$ values (Fig EV2A). As $n$ increases, we observe an increase in correlation (driven by the reduction in sampling noise), with a saturation beginning at $n = 20$. As saturating correlations may reflect the onset of blurred biological signals, we chose $n = 20$ for this analysis. We note that this selection also limits our ability to detect extremely rare transitions (< 0.1%) in the data.

In principle, averaging signals across single cells can potentially blend together signals from heterogeneous sub-populations. While we attempted to avoid this by only averaging cells in very similar transcriptional states, we wanted to ensure that our micro-clusters truly represented "homogeneous" populations. To do this, we tested whether dropout rates for genes within a micro-cluster were consistent with pure sampling noise. For each gene in each micro-cluster, we calculated the expected Poisson dropout rate (percentage of cells with zero detected molecules) based on its mean expression and compared this to the observed dropout rate (Fig EV2B and C). Overall, we observed very high correlations (0.98–0.99) between expected and observed dropouts, and this held across all micro-clusters (Fig EV2D, example shown for 100 randomly selected micro-clusters). This indicates that heterogeneity within a micro-cluster is driven primarily by sparse sampling as opposed to extensive biological heterogeneity, enabling us to pool information across cells in the same micro-cluster.

## Reconstructing developmental trajectories from micro-clusters

Upon the construction of a new dataset with micro-clusters, we sought to construct a developmental hierarchy based on gene expression. Given that a HSC can differentiate into cells of all possible lineages, we used the MST algorithm for hierarchical reconstruction. An MST seeks to find a subgraph that will span all the vertices, in this case micro-clusters, of a connected graph with the minimum sum of edge lengths. It has been previously applied to several trajectory-finding methods such as Monocle and SPADE (Qiu *et al*, 2011; Trapnell *et al*, 2014).

Prior to MST construction, we pre-processed our micro-cluster dataset using the same variable gene selection, normalization, and cell cycle regression strategy as with our original single-cell dataset. We reduced the dimensionality of this 5,000 × 963 micro-cluster profile using diffusion maps, implemented in the diffusionMap R package. We then constructed a distance matrix between micro-clusters, based on diffusion distance across 10 dimensions, although in practice we obtained very similar results even with as few as five dimensions. We chose an MST layout by computing t-Distributed Stochastic Neighbor Embedding (tSNE), run on the same distance matrix that was used for MST construction. Notably, the tSNE here is used only for visualization of the hierarchy. In Fig EV2E, we present an alternative visualization of the MST hierarchy, with a modified layout based on multidimensional scaling (MDS), that allows for easy visualization of the tree structure.

## Annotating transcriptional states on the reconstructed topology

The MST computed on diffusion distances represents an unrooted and multilineage developmental hierarchy. To parse this model, we first assigned a root node to the MST, choosing the micro-cluster with the highest expression of *AVP*, a gene most highly upregulated in our stem cell cluster (C6, Table EV1). We note our downstream results are highly robust to the exact choice of root, as long as we choose a root node corresponding to an HSC cluster. After root assignment, the "developmental progression" of each downstream node can be represented as the length of the shortest path connecting it to the root.

Next, we identified terminal leaves in the tree, representing nodes with only parents and no children. We calculated the path lengths from the root node to all terminal nodes and selected the four terminal nodes with the longest path length to represent the four "endpoints" for hematopoietic differentiation into distinct hematopoietic lineages. Each of these terminal nodes represented micro-clusters corresponding to distinct cell states as determined in Fig 1, specifically erythroid, eosinophil/basophil/mast, neutrophil/monocyte, and lymphoid progenitors. Therefore, we can treat the four terminal nodes as "endpoints" of developmental progression toward four distinct hematopoietic lineages.

We next identified the "branch points" in our proposed hierarchy, which can be directly determined from the MST structure. As described in the main text, the notation of an exact "branch point" represents a simplification of our data, but enables us to identify genes which are dynamic across the hematopoietic hierarchy. To identify these, we identified the shortest path along the MST between all pairs of terminal nodes. The point on each shortest path that is closest to the root node represents a transcriptomic bifurcation in the model.

Lastly, we assigned each "micro-cluster" a branch identity. To do this, we divided the MST into a series of "segments". These can be easily visualized in Fig EV2E. This figure shows the same MST structure as Fig 2B, but on a different layout, which is based on MDS of the MST-based distance matrix, ensuring that the different segments of the MST, and the "branch points" which connect them, can be easily visualized.

Cells located prior to the first bifurcation (the branch point closest to the HSC) are annotated as HSC/MPP, and segments leading to terminal nodes were named based on their downstream lineage (i.e., "Er", "Ba/Eo/Ma", "Lym", "Neu/Mo"). For intermediate segments, which were downstream of the first bifurcation but did not lead to terminal nodes, we assigned names based on lineage potential of cells downstream, including an EMP which gives rise to the first two lineages, and a lymphoid-primed multipotent progenitor (LMPP), based on previous knowledge of this cell state which can give rise to both lymphoid and select myeloid populations (Kohn *et al*, 2012).

## Bootstrapping developmental reconstruction

The MST process finds the path that connects all points in the dataset with minimum total length, representing a putative developmental trajectory through cellular "snapshot" data. While this and other graph-based strategies have been previously demonstrated to accurately reconstruct unidirectional and branching developmental processes (Bendall *et al*, 2011; Trapnell *et al*, 2014; Setty *et al*, 2016), the presence of "short-circuits", incorrectly drawn edges between cells in different developmental stages, can cause significant errors in this procedure. This concern is particularly relevant for MST construction, which shares similarities with single-linkage clustering. To ensure that our developmental reconstructions were not driven by these artifacts, we performed the MST-building process on 1,000 subsamples of our data (which each subsample containing 800 micro-clusters) and assessed the reproducibility across bootstraps. We used the same reconstruction procedure, consisting of MST construction based on diffusion map coordinates followed by branch annotation, for each subsample.

When assessing our bootstraps, we found that in 1,000 subsamples, we obtained identical hierarchical relationships as shown in Fig 2C. Therefore, we conclude that the hierarchical relationships we derive between HSC/MPP to the four downstream lineages are robust to potential artifacts in the MST-building procedure.

However, our megakaryocyte (Mk) micro-clusters (Fig EV2G) did not exhibit consistent relationships across bootstraps. We observed that Mk micro-clusters branched from different locations in the hierarchy in different subsamples, resulting in multiple potential models for Mk development (the relative proportion of subsamples leading to each model is shown in Fig EV2H). We therefore conclude that our dataset is insufficient to resolve the precise location of Mk branching and excluded this lineage from further analysis. Therefore, the hierarchy we propose in Fig 2C is consistent with the presence of a common progenitor for Mk and other lineages, but also with the potential for Mk to derive directly from HSC, as has been recently proposed (Grover *et al*, 2016).

To assess the reproducibility of our hierarchy with complementary methods, we took our 960 micro-clusters dataset (excluding early megakaryocyte progenitors) and applied theses as input to

Monocle (Trapnell *et al*, 2014; Qiu *et al*, 2017). ICA was used to reduce dimensions, and we ordered individual micro-clusters using the 596 branch-dependent genes, specifying num_paths = 4 in the function orderCells(). These results are visualized in Fig EV2F and suggest an identical developmental hierarchy to our observations.

## Identifying dynamic gene modules

Once we had a reconstructed developmental hierarchy of early hematopoiesis, we next asked how gene expression patterns varied across the lineages. For example, Fig 2D exhibits expression patterns for canonical markers, revealing that many key hematopoietic regulators significantly diverge in their expression at fate transitions.

We therefore designed a test to identify, in an unsupervised way, branch-dependent genes whose expression levels were dynamic across any of the bifurcations in our model. For each "branch point", we performed the following test. We considered all nodes downstream of a "branch point", partitioning them into two groups based on the initial bifurcation. We then linearly scaled (normalized) the "developmental progression" for nodes along the "left" branch to fall between −1 and 0, and nodes along the right branch to fall between 0 and 1. A normalized value of −1 or 1 indicates a node which is farthest from the root, on either the left branch or right branch. Effectively, this normalization step allows the divergence on both branches to receive equal weight in the downstream test. Therefore, for each branch point, we obtained a vector $\mathbf{v}$, which contained the normalized developmental progression for all downstream micro-clusters, ranging between −1 and 1.

Branch-dependent genes whose expression bifurcates at the branch point should therefore have expression levels that are strongly correlated (or anti-correlated) with vector $\mathbf{v}$. Therefore, for each gene $g$, we computed a branch score Branch Score $(g) = \mathbf{x} \cdot \mathbf{v}$, where $\mathbf{x}$ represents the $z$-score for $g$ in each micro-cluster.

We observed that branch scores across all genes roughly obeyed a normal distribution, as the majority of genes across the transcriptome were not branch-dependent. We therefore selected positive and negative outlier genes whose branch score was greater than 2.5 times standard deviations from the mean of the distribution. Across all three "branch points", we detected a total of 596 branch-dependent genes. Lastly, we grouped genes into modules with similar developmental dynamics using $k$-means clustering with 100 random starts, using the kmeans() function. After clustering, we identified two groups of 40 and 39 genes that were primarily very lowly expressed, exhibited poor within-cluster similarities, and no hematopoietic ontology enrichments, and therefore removed these genes from further analysis. Clustering results, including all removed genes, are shown in Table EV3.

## Aligning datasets from human bone marrow and cord blood

The raw scRNA-seq read counts for human bone marrow CD34$^+$ Quartz-seq cells were downloaded from NCBI GEO (GSE75478). To integrate this dataset with our Drop-seq micro-clusters, we ran the scRNA-seq integration procedure as described in Seurat 2.0 (Satija *et al*, 2015; preprint: Butler & Satija, 2017). Briefly, the procedure aims to identify potentially shared subpopulations between two datasets, based on shared sources of variation. We identified the top variable genes in each dataset (quantifying dispersion as a

variance/mean ratio) with the default parameters and used the union of these two gene sets as input to the procedure.

We first learned the common sources of biological variations between datasets by performing a canonical correlation analysis (CCA). The canonical correlation vectors (CCs) 1–7 were then used as the "scaffolds" for alignment with a non-linear "warping" approach implemented in the AlignSubspace function. These aligned CCs were used as input for co-clustering, again with modularity optimization from a shared nearest-neighbor (SNN) graph to identify shared subpopulations. The co-clustering returned subpopulations that were consistent with our independent analysis of cord blood micro-clusters, as well as the analysis in the original bone marrow study (Velten *et al*, 2017). To annotate cells in both dataset, we assigned all cells in a co-cluster to the micro-cluster branch ID with maximal membership in the cluster. We also evaluated the robustness for alignment by randomly sampling 500 cells from the bone marrow dataset and repeated the alignment to the full cord blood micro-clusters data. The results between the annotations in the subsampled and full analysis were highly consistent, particularly for the "endpoint" clusters, and are shown in Fig EV4E.

To compare the expression dynamics between the two systems, we applied a scoring method from Tirosh *et al* (2016), assigning each gene module an "expression score" within every cell. Briefly, we grouped all genes into 25 bins according to aggregated expression levels and selected 100 "control genes" from the same bin as a gene from the analyzed gene set. The score was then calculated by subtracting the average expression of the gene set by the aggregated values of the control gene set, to control for differences in the complexity and dropout rates across single cells.

## ATAC-seq analysis

The count matrix for ATAC-seq profiles of hematopoietic and leukemic cell types (132 samples in total) was downloaded from NCBI Gene Expression Omnibus (GSE74912) (Corces *et al*, 2016). Peaks were quantile-normalized using the normalize.quantiles() function in R package preprocessCore. We also scaled the peaks between 0 and 1 using the rescale() function in the R scales package, clipping at 5 and 95% quantiles for every peak across samples. Each peak was associated with a nearby TSS using annotatePeaks.pl from HOMER (Heinz *et al*, 2010), with human hg19 as a reference. To filter out peaks with low accessibility, we calculated the maximum normalized signals across samples (we selected samples from the following cell type: CLP, GMP, CMP, HSC, LMPP, MEP, MPP), and removed peaks with a maximum value less than 80 from downstream analysis.

To define the variable loci, we calculated the mean and standard deviation for every peak, and selected the top 2,000 peaks with the highest coefficient of variation (CV, standard deviation divided by the mean) and performed PCA to learn the primary structure in early hematopoietic regulation. To retrieve the "primary peak" per gene, we compared the range of normalized signals for peaks associated with the same gene across all cell types and used the one with the maximal accessibility as the primary peak. The width of the "river" plots used in Figs 4G and EV4G represents the mean width of all primary peaks in each gene module.

To visualize modules of ATAC-seq peaks with similar dynamic patterns, we used constrained *k*-means clustering on peaks assigned to a dynamic gene module (e.g., "*de novo* lymphoid" genes), setting

$k = 4$ and $\alpha = 0.2$. To systematically group loci into "consistent" and "inconsistent" types, we leveraged the ranking of peaks associated with one gene module across different cell types. For each cell type, we averaged the normalized signals per accessible region across samples and ranked the averaged signals among MEP, CMP, MPP, HSC, LMPP, GMP, and CLP. Peaks with highest ranks in the consistent cell type (e.g., peaks assigned to "*de novo* lymphoid" genes with the highest rank in CLP) were defined as being "consistent", whereas other peaks ("*de novo* lymphoid" genes with peaks highest in MEP) were defined as being "inconsistent".

For each gene module shown in Fig 3A, the genomic positions of either consistent or inconsistent peaks were used for motif enrichment, using the findMotifsGenome.pl command in HOMER, with hg19 as the reference genome and the default settings for all other options. To visualize the shared motifs from different peak classifications, we combined the top 30 motifs of each group to form a list for potential transcriptional regulators. The negative log *P* values corresponding to these motifs were retrieved from HOMER output, and we took those with high enrichment (maximum $-\log P > 10$ in at least one peak classification) for visualization in heatmaps using heatmap.2() in gplots.

## Coupling transcriptomic data with cellular immunophenotypes

To evaluate the surface immunophenotypes for Drop-seq clusters, we sequenced the transcriptomes for 96 mast cell progenitors (CD34$^+$ CD117$^+$ FcεRIα$^+$) and 865 canonically defined LMPPs (CD34$^+$ CD38$^-$ CD45RA$^+$) from indexed FACS sorting. The raw data processing and quality control were the same as described for Drop-seq data. To associate FACS-sorted cells with cells profiled with Drop-seq, we leveraged information from the reconstructed hierarchy with micro-clusters by building a random forest classifier. Briefly, we assigned a branch identity to each single cell from Drop-seq, based on its corresponding micro-cluster. A random forest classifier was then trained on these cells using the ranger() function in R with default parameters. We used the 517 "branch-dependent" genes for classifier construction. We then applied the classifier on FACS-sorted cells to reveal the transcriptomic states and compared protein signals for surface markers with indices recorded on the sorter (CD10, CD52, CSF3R, CD38, CD45RA and CD62L for LMPP, and CD71 for mast cell progenitors).

### *In vitro* differentiation with MS5-MBN assay

For our *in vitro* differentiation experiments, we utilized the differentiation protocol described in (Laurenti *et al*, 2013). Mouse MS5 stromal cells were treated with 10 μg/ml mitomycin C (in alpha MEM media) for 3 h to inhibit mitotic proliferation, and seeded on 0.2% gelatin-coated 96-well cell culture plates 24 h before human cell sorting. Approximately $4 \times 10^4$ MS5 cells were plated per well in H5100 media (Stem Cell Technologies). Human CD34$^+$ cells were enriched from three separate healthy cord blood units as described above, using Ficoll density centrifugation (GE Healthcare) and MACS magnetic separation (Miltenyi). Each cord blood unit was processed individually; 24 h after plating MS5 cells, we replaced the media with fresh H5100 supplemented with the following cytokines (R&D systems): SCF (10 ng/ml), Flt3L (10 ng/ml), TPO (50 ng/ml), IL-2 (10 ng/ml), IL-7 (20 ng/

ml), IL-6 (20 ng/ml), G-CSF (20 ng/ml), and GM-CSF (20 ng/ml). Enriched $CD34^+$ cells were stained with the following antibodies: APC mouse anti-human CD34 (BD #560940, clone 581), Alexa Fluor 700 mouse anti-human CD38 (BD #560676, clone HIT2), APC/Cy7 anti-human CD45RA antibody (BioLegend #304127, clone HI100), CD52 monoclonal antibody FITC (Thermo Fisher Scientific #MA1-82037, clone HI186), BV421 mouse anti-human CD10 (BD #562902, clone HI10a), and PE mouse anti-human CD114 (CSF3R, BD #554538, clone LMM741), with concentrations determined from titrations, and 250–300 cells ($CD34^+$ $CD38^-$ $CD45RA^+$ $CD10^-$ $CD52^+$ $CSF3R^-$, or $CD34^+$ $CD38^-$ $CD45RA^+$ $CD10^-$ $CD52^-$ $CSF3R^+$) were sorted onto stromal cells. Cell culture was maintained in $CO_2$ incubator for 3 weeks, with weekly change of half the cytokine-supplemented media. At the end of the third week, cells were harvested by pipetting and stained with the following antibodies: Alexa Fluor® 700 anti-human CD45 antibody (BioLegend #304023, clone HI30), PE Mouse Anti-Human CD56 (BD #556647, clone MY31), APC/Cy7 Mouse Anti-Human CD11b (BD #560914, clone ICRF44), and FITC Mouse Anti-Human CD19 (BD #555412, clone HIB19) and resuspended in MACS buffer (PBS + 0.5% BSA + 2 mM EDTA) with DAPI before flow cytometry. Live human cells were identified as $DAPI^-$ $hCD45^+$, and differentiation output was evaluated as $CD19^+$ $CD11b^-$ B cells, $CD56^+$ $CD11b^-$ NK cells and $CD56^-$ $CD11b^+$ myeloid cells within the live human cell gate. Populations were gated on FlowJo and cell type proportions were analyzed in R. We calculated the lymphoid (B and NK cells) to myeloid ratio for each replicate per unit, and normalized by the average lymphoid to myeloid ratio from $CD52^+$ LMPPs per unit.

## Data availability

Raw fastq files $CD34^+$ Drop-seq and plate-based scRNA-seq data, and digital expression matrix for Drop-seq are available in NCBI Gene Expression Omnibus with the primary accession code GSE97104. An online resource to visualize the expression levels of any user-defined gene along our reconstructed trajectories is also available at http://satijalab.org/cd34/.

Expanded View for this article is available online.

## Acknowledgements
We acknowledge C. Hafemeister, L. Wu, A. Powers, and L. Harshman for critical discussions and support. This study was supported by funds from the New York Genome Center.

## Author contributions
SZ and RS designed the study; SZ, EP and WS performed the experiments; SZ, AB and RS analyzed the data; and SZ and RS wrote the manuscript.

## Conflict of interest
The authors declare that they have no conflict of interest.

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
