## [Review Process File · Molecular Systems Biology]

Molecular transitions in early progenitors during human cord blood hematopoiesis

Shiwei Zheng, Efthymia Papalexou, Andrew Butler, William Stephenson and Rahul Satija

Review timeline:

Submission date:	9 October 2017
Editorial Decision:	18 February 2018
Revision received:	7 February 2018
Accepted:	12 February 2018

Editor: Maria Polychronidou

Transaction Report:

1st Editorial Decision

18 February 2018

Thank you again for submitting your work to Molecular Systems Biology. We have now heard back from the two referees who agreed to evaluate your study. As you will see below, the reviewers are overall quite positive and think that the findings seem interesting. They raise however a series of concerns, which we would ask you to address in a revision of the manuscript.

The reviewers' recommendations are rather clear and therefore I think that there is no need to repeat all the points listed below. Please let me know in case you would like to discuss any of the issues raised by the reviewers.

REVIEWER REPORTS

Reviewer #1:

In this paper the authors apply large-scale scRNA-seq profiling based on Drop-seq to characterize the transcriptional diversity and lineages identified in CD34+ human cord blood cells. Overall, this study is interesting and unique in scale and approach. The analysis of primary cord blood from multiple donors represents a significant advance and complements previous analyses of lineages in human bone marrow. My concerns are two-fold.

First, the authors should work on the text to clearly mark and highlight novel findings and insights derived. As presented, the analysis feels fairly descriptive and mostly confirmatory. Second, I have some technical concerns and comments that require additional analyses.

Specific comments:

1. Robustness of the clustering & outlying cells.

Due to the central use of the clustering step, it would be helpful to confirm the determined clustering using alternative strategies and methods. I would also be interested to see more data about the outlying cells that are discarded. Can these cells be aligned to intermediates states in the reconstructed graphs or these truly rare / low quality / noise ?

2. Biological relevance and completeness of the identified micro clusters.

The authors make the interesting remark that the variation within these clusters is consistent with Poisson noise. I would suggest to stress this message, but also add additional controls to clarify whether the N~900 micro clusters do indeed represent the transcriptional complexity of the system. In particular, because transcriptome profiles have been used to define the clusters, there is the risk of overfitting. To address this, I would suggest a hold-out procedure, thereby demonstrating that genes that were excluded during the definition of clusters retain Poisson-like variability between cells within the clusters.

3. Extension of the transcription factor motif analysis.

The motif analysis for genes in distinct modules across lineages (page 9) is interesting and offers the opportunity to obtain mechanistic insights. I would suggest to extend this analysis. For example, it would be interesting to understand whether motifs are predictive of fine-grained differences of trajectories for individual genes, e.g. using known targets of TFs.

4. Technical controls for the bone marrow data integration.

The integration of scRNA-seq from this study with existing data from bone marrow is interesting. As these alignment methods are still a fairly recent development, I would request additional technical controls to show that the (impressive!) agreement between studies is not the results of overfitting. E.g. can the method be run in hold out manner, using on a subset of cells and/or genes, to confirm the robustness of the mapping between studies?

5. ATAC-se integration.

This is the most descriptive part of the paper and I find the insights appear to be rather slim. It would be helpful to work out any messages more clearly. From my perspective the section could also be toned down/dropped.

Reviewer #2:

Zheng et al report the generation and comprehensive analysis of a single cell gene expression dataset for human cord blood CD34 positive cells, which comprises a broad stem/progenitor mix of human blood cells. The authors identify 4 distinct "endpoints" of maturing cells, reveal intermediate differentiation stages that show evidence of multi-lineage priming, explore the relationship between chromatin state and "transcriptomic differentiation state", and carry out single cell functional assays that exploit - and then validate - predicted heterogeneity within the putative LMPP compartment. The study is on the whole well executed, and the conclusions supported by the data. However, there are a few specific areas where the paper could be improved, as outlined below:

Specific Comments

1) I would argue that the potential impact of this paper could be greatly enhanced if the authors provide a user-friendly website that would allow the wider scientific community to explore and download the data. I am not asking for a website that would run analysis, just something simple as was provided for the Nestorowa mouse scRNA-Seq paper that they cite. In addition, I could also not see a link to accession numbers in the main document, which would need to be provided too.

2) Page 4: The authors provide the number of UMIs per cell, but should also state here the number of detected genes per cell. This is important bit of information for the community, when reading a given paper, and thinking about how datasets relate to each other.

3) Still page 4: The authors should also say something here about the expected rate of doublets, and whether or not they have done something bioinformatically to lessen their impact on subsequent data analysis.

4) Page 6: The authors need to justify why mini clusters of 20 cells is a good number. What happens with 10 cells, what happens with 25 or 50 cells?

5) Still about the miniclusters: Does the minicluster analysis in some sense mean that the dataset shrinks from 20,000 to 1,000 entities? Because this is in the same range of cells analysed by the Velten et al paper by the deeper-sequencing scRNA-Seq method.

6) Why does the diffusion map in figure 2D not reveal the 4 endpoints? Would they be seen when looking at further dimensions? It may be worth commenting on this. And more generally, whether the tree hierarchy was also seen when using alternative methods of data analysis (there are quite a few now for finding branched differentiation trajectories in single cell data).

Minor Points:

1) Page 1: Although the term "pluripotent" used to be widely used for HSCs (and of course when translated into English does capture what they do), it is these days almost exclusively used for embryonic stem cells. Would be better therefore to use multipotent.

2) Figure 2B/D: It would be good to remove the black outlines of the circles, and then use colors to differentiate high/low expression. I had to zoom in really high on my computer to see the expression in panel D.

We appreciate the positive comments from the two peer reviewers, and were gratified to see their enthusiasm for our work. We recognize that both of the reviewers had additional questions regarding the robustness and reproducibility of some of our approaches. We have revised our manuscript in line with these concerns, and describe our modifications in the response below. We believe that these have strengthened the overall work, and thank the reviewers for their constructive comments.

Reviewer #1: In this paper the authors apply large-scale scRNA-seq profiling based on Drop-seq to characterize the transcriptional diversity and lineages identified in CD34+ human cord blood cells. Overall, this study is interesting and unique in scale and approach. The analysis of primary cord blood from multiple donors represents a significant advance and complements previous analyses of lineages in human bone marrow. My concerns are two-fold. First, the authors should work on the text to clearly mark and highlight novel findings and insights derived. As presented, the analysis feels fairly descriptive and mostly confirmatory. Second, I have some technical concerns and comments that require additional analyses.

We thank the reviewer for these constructive comments, which we address below.

Specific comments:

1. **Robustness of the clustering & outlying cells.** Due to the central use of the clustering step, it would be helpful to confirm the determined clustering using alternative strategies and methods. I would also be interested to see more data about the outlying cells that are discarded. Can these cells be aligned to intermediates states in the reconstructed graphs or these truly rare / low quality / noise?

We agree with the reviewer that the robustness of our clustering is an important area to explore further. Our approach of dimensional reduction followed by graph-based clustering is the most widely used clustering approach in the field, but in our revised manuscript, we asked whether our results were not dependent on specific key parameters that can alter the results. To address this, we conducted a robustness analysis of our clustering results, where we ran the clustering over a range of values for two key parameters: the number of nearest neighbors, and the clustering resolution/granularity. Overall, we ran 25 different clustering, and combined the results into a consensus clustering matrix, which allowed us to calculate how often each pair of cells clustered together across the 25 re-clusterings.

We observed that cell pairs that clustered together in the original analysis tended to cluster together in the >80% of the re-clusterings. This was particularly striking for the more differentiated cells (across the four 'endpoints' clusters), where this frequency was 92%, as the boundaries are more clearly defined between cell states as expected. We conclude that our original clustering is a faithful representation of

the data and is not tuned to particular parameter values. We thank the reviewer for this suggestion, and include these analyses in Extended View Figure 1C.

The reviewer also asked for further clarification of the rare cells that we excluded from downstream analysis. As shown in Extended View Figure 1B, these clusters were highly enriched for mRNA markers that are canonically associated with differentiated cell markers, including CD3D (T cells), MS4A1 (B cells), and C5AR1 (neutrophils). Additionally, these cells were depleted of transcripts for stem cell/progenitor markers (for example, *GATA2*, *KIT*, *FLT3* or *CSF3R*). Taken together, we are confident that the outlying cells are not intermediate states in the reconstructed trajectories, but are likely CD34^{-/low} cells that inadvertently passed through column purification, and should therefore be excluded from downstream analyses.

2. Biological relevance and completeness of the identified micro clusters. The authors make the interesting remark that the variation within these clusters is consistent with Poisson noise. I would suggest to stress this message, but also add additional controls to clarify whether the N~900 micro clusters do indeed represent the transcriptional complexity of the system. In particular, because transcriptome profiles have been used to define the clusters, there is the risk of overfitting. To address this, I would suggest a hold-out procedure, thereby demonstrating that genes that were excluded during the definition of clusters retain Poisson-like variability between cells within the clusters.

We agree with the reviewer and have slightly expanded this section, with two new analyses. First, we examined all 31,289 genes that were not included in our clustering ('hold-out' analysis), and examined their variation levels. These analyses are shown in Extended View Figure 2B, and we conclude that even genes that were not involved in clustering also exhibit Poisson noise between single cells in a micro-cluster. Second, we justify our choice of n = 20 for pooling micro-clusters with a saturation analysis shown in Extended View Figure 2. Larger values of n introduce additional smoothing, but with diminishing returns, and creating a risk of blurring biological distinctions. Together, these analyses address the reviewer's concerns about potential overfitting.

3. Extension of the transcription factor motif analysis. The motif analysis for genes in distinct modules across lineages (page 9) is interesting and offers the opportunity to obtain mechanistic insights. I would suggest to extend this analysis. For example, it would be interesting to understand whether motifs are predictive of fine-grained differences of trajectories for individual genes, e.g. using known targets of TFs.

5. ATAC-seq integration. This is the most descriptive part of the paper and I find the insights appear to be rather slim. It would be helpful to work out any messages more clearly. From my perspective the section could also be toned down/dropped.

We thank the reviewer for the constructive comments. We realized that the ATAC-seq data presented an opportunity to extend the motif analyses, as the reviewer suggests, to better understand how differences in motif content correlate with transcriptional dynamics. The analyses presented in our initial manuscript focused only on the global chromatin dynamics of each gene, a summary of each gene's accessibility that considered the effect of only a single accessible region for each gene.

In our revised manuscript, we have included a broader investigation on the full dataset with all the open accessible regions detected. This revealed a set of intriguing peaks, which we deem 'inconsistent' peaks, that exhibit opposing accessibility dynamics compared to the transcriptional output of the nearest gene. These peaks are decidedly in the minority (~15% of all peaks), and contain strikingly different motif enrichments compared to "consistent" peaks, despite being located upstream of the same genes. This is clearly exhibited with an example at the CSF3R promoter in Figure 4H, alongside additional downstream analysis in Figure 4I and Extended View Figure 4.

We were surprised by this degree of heterogeneity in chromatin accessibility dynamics in a single promoter. While we emphasize that these inconsistent peaks are in the minority, and may have no function on transcriptional output, they could also represent intriguing cases of "cross-antagonism". In this case, these peaks potentially serve as repressive binding sites, keeping genes repressed after down-regulation. While such examples have been reported anecdotally for ETS and GATA binding factors, repressive regions are expected to play important roles across the transcriptome, consistent with our findings here.

Due to the extreme technical challenge of manipulating or perturbing regulatory regions in primary human hematopoietic cells, this analysis remains descriptive. However, we believe that the identification of these regions, along with a detailed description of their motif content, represents a potentially valuable insight that can result from the integration of scRNA-seq and ATAC-seq data.

4. Technical controls for the bone marrow data integration. The integration of scRNA-seq from this study with existing data from bone marrow is interesting. As these alignment methods are still a fairly recent development, I would request additional technical controls to show that the (impressive!) agreement between studies is not the results of overfitting. E.g. can the method be run in hold out manner, using on a subset of cells and/or genes, to confirm the robustness of the mapping between studies?

We appreciated that the reviewer found these analyses interesting and agreed that their robustness could be further explored. To address this concern, we performed a repeated subsampling analysis, where we sampled 500 cells from the bone marrow dataset, and aligned them to the cord blood micro-clusters as we did with the full

Velten dataset. Visualized as a 'confusion matrix' in Extended View Figure 4, we found that the alignment results are consistent between the 500-cell subset (median consistency 'on-diagonal' of 0.70) and all bone marrow cells, as shown in the consensus matrix from Extended View Figure 4E. In the cases where we observed differences, this was largely driven by blurred cell state boundaries in early intermediate states (for example, the exact cutoff between HSC/LMPP), as would be expected for imposing clustering onto a continuous process. As expected, we also observed higher values for 'endpoint' clusters (median consistency 0.85). Again, this addresses the possible concern for overfitting, and we thank the reviewer for this suggestion.

Reviewer #2: Zheng et al report the generation and comprehensive analysis of a single cell gene expression dataset for human cord blood CD34 positive cells, which comprises a broad stem/progenitor mix of human blood cells. The authors identify 4 distinct "endpoints" of maturing cells, reveal intermediate differentiation stages that show evidence of multi-lineage priming, explore the relationship between chromatin state and "transcriptomic differentiation state", and carry out single cell functional assays that exploit - and then validate - predicted heterogeneity within the putative LMPP compartment. The study is on the whole well executed, and the conclusions supported by the data. However, there are a few specific areas where the paper could be improved, as outlined below:

Specific Comments 1) I would argue that the potential impact of this paper could be greatly enhanced if the authors provide a user-friendly website that would allow the wider scientific community to explore and download the data. I am not asking for a website that would run analysis, just something simple as was provided for the Nestorowa mouse scRNA-Seq paper that they cite. In addition, I could also not see a link to accession numbers in the main document, which would need to be provided too.

We have created a webpage, based on an R 'Shiny' app, to help visualize gene expression in our reconstructed trajectories. This is openly and freely available at : <http://www.satijalab.org/cd34/>. In addition, the app allows for the visualization of the integrated bone marrow and cord blood datasets, as well as gene expression levels in the Laurenti et al. microarray dataset. Lastly, our data is uploaded to NCBI Geo with the accession number of GSE97104, and the token for reviewer access is 'evedicoslnyrdeh'. We have listed these resources in a 'data availability' section at the end of the manuscript.

2) Page 4: The authors provide the number of UMIs per cell, but should also state here the number of detected genes per cell. This is important bit of information for the community, when reading a given paper, and thinking about how datasets relate to each other.

We agree and thank the reviewer for the suggestion, and have included this information in the main text (1,046 genes detected per cell, 6,858 genes per micro-cluster on average).

3) Still page 4: The authors should also say something here about the expected rate of doublets, and whether or not they have done something bioinformatically to lessen their impact on subsequent data analysis.

We appreciate the reviewer's suggestion – especially as doublet states could appear to represent intermediate populations in our data. This concern is primarily relevant for extremely rare intermediates.

In our optimization of the Drop-seq technology, we chose to use cell and microparticle flow rates that yielded expected doublet rates of 1-2%. Therefore, if there were intermediate populations of this rarity in the data, we would agree with the reviewer's concern. However, given the abundance of intermediate clusters (ranging from 18-23%), we are fully confident that these clusters cannot be a byproduct of cell doublets.

While we could choose to exclude cells with higher numbers of UMIs as putative doublets, we worried that we may introduce bias against larger cells into our downstream analyses. Therefore, we chose to keep with existing analysis in the field (including Velten et al., Paul et al., and Nestorowa et al.) and not attempt to bioinformatically detect and remove doublets.

We thank for reviewer for raising this, and have now included our expected doublet rate in the main text and Materials and Methods.

4) Page 6: The authors need to justify why mini clusters of 20 cells is a good number. What happens with 10 cells, what happens with 25 or 50 cells?

We appreciate the reviewer's concern and have provided additional analysis shown in Extended View Figure 2D. As suggested, we varied the number of cells pooled together in microclusters, and computed downstream technical metrics to justify our choice of $n = 20$. In particular, computed the correlation and covariance between two neighboring micro-clusters after the averaging. As we show in the new Figure, the correlation values increase and eventually reach a saturation as more single cells are included in one micro-cluster, indicating the increasing degree of smoothness for our dataset, while >0.9 correlation is reached when $n = 20$. Meanwhile, the covariance between two nearest neighbors starts to drop as more cells are included, which suggests a decrease on resolution. Therefore, we chose $n = 20$ to reach a balance between smoothness and resolution. We note that we obtain the same global biological hierarchies with slightly different values of n , but hope that these new analyses justify our choice of this parameter.

5) Still about the miniclusters: Does the minicluster analysis in some sense mean that the dataset shrinks from 20,000 to 1,000 entities? Because this is in the same range of cells analysed by the Velten et al paper by the deeper-sequencing scRNA-Seq method.

Indeed, as the reviewer suggests, performing micro-clustering does reduce the number of cells in our data. However, we gain a significant boost in sensitivity, even compared to deep single cell RNA-seq technologies, as applied in the Velten et al. Manuscript. In particular, we have observed a striking increase in detected genes/cell per microcluster (6,858 genes per micro-cluster, compared with 3,758 genes from the Velten et al. manuscript). The increase in gene numbers help us better identify the top enriched markers for each progenitor states, as visualized from the side-by-side heatmaps in Figure 4C.

6) Why does the diffusion map in figure 2D not reveal the 4 endpoints? Would they be seen when looking at further dimensions? It may be worth commenting on this. And more generally, whether the tree hierarchy was also seen when using alternative methods of data analysis (there are quite a few now for finding branched differentiation trajectories in single cell data).

We apologize for the confusion; Figure 2D does in fact reveal all four endpoints, and the layout in Figure 2D is identical to Figure 2B, which contains the annotation for each progenitor state. The four endpoints are those labeled as 'Ba/Eo/Ma', 'Er', 'Neu/Mo' and 'Lym'. We have added the explanation in the figure legend.

We also agree that our observed hierarchy should be reproducible with other tools analyzing single cell trajectories. To address this, we have run Monocle on our micro-clusters. As shown in Extended View Figure 2H, Monocle reveals the same tree hierarchy with four 'endpoints' – Ba/Eo/Ma, Er, Neu/Mo and Lym. We therefore conclude that similar biological results can be obtained by running multiple analytical tools.

Minor Points: 1) Page 1: Although the term "pluripotent" used to be widely used for HSCs (and of course when translated into English does capture what they do), it is these days almost exclusively used for embryonic stem cells. Would be better therefore to use multipotent.

2) Figure 2B/D: It would be good to remove the black outlines of the circles, and then use colors to differentiate high/low expression. I had to zoom in really high on my computer to see the expression in panel D.

We have made these modifications as requested.

Thank you for sending us your revised manuscript. We are now satisfied with the modifications made and I am pleased to inform you that your paper has been accepted for publication.

Corresponding Author Name: Rahul Satija

Manuscript Number: MSB-17-8041